# Comparative 3D ultrastructure of *Plasmodium falciparum* gametocytes

Felix Evers [1], Rona Roverts [2,3], Cas Boshoven [1], Mariska Kea-te Lindert[2,3], Julie M. J. Verhoef [1], Nico Sommerdijk [2,3], Robert E. Sinden[4], Anat Akiva [2,3] & Taco W. A. Kooij [1] ✉

Despite the enormous significance of malaria parasites for global health, some basic features of their ultrastructure remain obscure. Here, we apply high-resolution volumetric electron microscopy to examine and compare the ultrastructure of the transmissible male and female sexual blood stages of *Plasmodium falciparum* as well as the more intensively studied asexual blood stages revisiting previously described phenomena in 3D. In doing so, we challenge the widely accepted notion of a single mitochondrion by demonstrating the presence of multiple mitochondria in gametocytes. We also provide evidence for a gametocyte-specific cytostome, or cell mouth. Furthermore, we generate the first 3D reconstructions of the parasite's endoplasmic reticulum (ER) and Golgi apparatus as well as gametocyte-induced extraparasitic structures in the infected red blood cell. Assessing interconnectivity between organelles, we find frequent structural appositions between the nucleus, mitochondria, and apicoplast. We provide evidence that the ER is a promiscuous interactor with numerous organelles and the trilaminar pellicle of the gametocyte. Public availability of these volumetric electron microscopy resources will facilitate reinterrogation by others with different research questions and expertise. Taken together, we reconstruct the 3D ultrastructure of *P. falciparum* gametocytes at nanometre scale and shed light on the unique organellar biology of these deadly parasites.

Parasites from the genus *Plasmodium* are the causative agents of malaria. This mosquito-borne infectious disease remains a huge burden on global public health with more than 200 million cases and > 600 thousand fatalities in 2021 alone[1]. Humanity's efforts to combat this disease have shown impressive results during the years 2000 – 2015 but have since stalled. This is largely driven by continuing emergence of resistance to all frontline antimalarials[2] as well as the inability of most antimalarials to combat directly the asymptomatic but transmissible sexual stages[3]. To stop parasite transmission and eliminate malaria, there is an urgent need for drugs with novel mechanisms of action, particularly those that are effective against

sexual stages. The apicomplexan phylum, among which are the malaria parasites, and other related single-cell eukaryotes diverged very early in evolution from more commonly studied species including plants, yeast, animals and humans[4–6]. Consequently, various basic cellular processes and structures are comparatively poorly understood in Apicomplexa despite their immense significance for global health. In particular, *Plasmodium* organelle biology has been shown to deviate from standard eukaryotic models and varies drastically between the different life-cycle stages[7,8]. Furthermore, Apicomplexa are almost exclusively parasitic, a strategy that drives additional divergence in cellular features related to their survival strategy, such as feeding,

[1]Department of Medical Microbiology, Radboud University Medical Center, Nijmegen, The Netherlands. [2]Electron Microscopy Center, RTC Microscopy, Radboud University Medical Center, Nijmegen, the Netherlands. [3]Department of Medical Biosciences, Radboud University Medical Center, Nijmegen, the Netherlands. [4]Department of Life Sciences, Imperial College London, London, UK. ✉e-mail: taco.kooij@radboudumc.nl

locomotion, secretion, invasion, and adaptation to their respective hosts. As a result, we are in the situation where there is urgency to find drug or vaccine candidates for cellular systems that are relatively poorly understood. In the case of *Plasmodium*, this holds particularly true for life-cycle stages, like the sexual gametocytes, that are less accessible than the pathogenic asexual blood-stage parasites (ABS).

Classical electron microscopy studies in the period from 1965 to 2000 provided key insights into the ultrastructure of malaria parasites and laid the groundwork for much of our understanding of sexual-stage biology[9–12]. There have been a few noteworthy serial sectioning electron microscopy (ssEM) applications[13–16], however, these early studies lacked the advanced quantitative, volumetric, and computational methods available today and did not have the context of our current molecular understanding of malaria parasite biology. With the advent of novel and, more importantly, increasingly accessible high-resolution volumetric approaches such as focused ion beam milling - scanning electron microscopy (FIB-SEM), serial block-face scanning electron microscopy (SBF-SEM), array tomography (AT), and expansion microscopy (ExM), great advances have been made in recent years. Features of ABS and oocysts and their respective replication strategies have been investigated via FIB-SEM with great success[17,18]. ssEM and AT have recently been applied to elucidate the reputed role of nuclear microtubules in driving the characteristic elongation of developing gametocytes[19] and general measurements of gametocytes and insights underpinning the inner membrane complex (IMC) have been gained using SBF-SEM[20]. ExM has been used to comprehensively map asexual blood-stage development[21] and to elucidate microtubule dynamics across gametocyte development and during activation, and immunofluorescence-based studies have suggested that the mitochondrion dramatically branches and enlarges in the gametocyte stages[22–24]. Additionally, electron micrographs have shown that in contrast to the acristate mitochondrion in ABS, the gametocyte mitochondrion is highly cristate[10,25,26]. Yet, other organelles, such as the endoplasmic reticulum (ER), the Golgi apparatus (Golgi), and the cytostome, and their interconnectivity have not been subject to specific microscopic investigations in gametocytes and our general understanding of their morphology and putative interactions is largely derived from model eukaryotes or extrapolated from *Plasmodium* ABS.

In this study, we investigate the ultrastructure of gametocytes using nanometre resolution FIB-SEM and ssEM volumetric imaging. We compare our ultrastructural gametocyte findings with our ABS data and other high-resolution volumetric studies. In doing so, we visualize the 3D morphology and distribution of surprisingly abundant external structures in gametocytes and find evidence for widespread inter-organelle interactions. Furthermore, we identify a morphologically distinct cytostome with extensive membrane folds deviating from the classical appearance in ABS of this so-called cell mouth. Finally, we demonstrate the presence of multiple mitochondria in a single gametocyte, breaking a decades old dogma that malaria parasites would have a single copy throughout all stages of development. In conclusion, we generate high-resolution renderings of various organelles shifting our understanding of gametocyte morphology. Thus, we created a reusable image resource (deposited on EMPIAR[27], EMPIAR-12160) for the malaria research community to investigate more cells from the FIB-SEM and ssSEM data, contrast their own findings with a reference point, and reinterrogate the datasets with their specific biological question, perspective, and expertise, such as we recently did when focusing on organelle segregation during schizogony[28].

## Results and Discussion
### General morphology of asexual and sexual blood-stage parasites
The mature gametocytes show their typical crescent shape that is well known from light microscopy-based imaging (Fig. 1A, B, Movies S1, 2).

The surrounding red blood cell (RBC) also conformed to this general shape, with Laveran's bib, a narrow ridge spanning between the tips of the parasite crescent, being evident in all stage IV and V gametocytes. We also identified contorted mature gametocytes with a twisted appearance including a narrowing of the cytoplasm, a frequent aberrant morphology known from Giemsa preparations of gametocyte cultures (Fig. S1). We were surprised to find that aside from the deviating morphology, no other signs of poor health or differences of internal structures could be recognized in these contorted gametocytes. Immature gametocyte stages (stage II-IV) were identified based on their well-characterized straighter or more compact appearance and incomplete inner membrane complex (Fig. S2). In more mature gametocytes, the overall electron density of the RBC cytoplasm decreases and a particular translucent corona appears around the gametocyte, indicative of haemoglobin depletion (Fig. S3). While we were unable to confidently assign sexes to earlier stages, we relied on differing patterns of haemozoin distribution, the frequency of osmiophilic bodies, nuclear shape, and prevalence of ER to differentiate male and female stage IV/V gametocytes as reported previously[10]. Whereas previously cytoplasmic density of ribosomes[29] was widely used to discriminate sexes, we could not detect such differences in our preparations. Appearance of nucleus, ER, Golgi, mitochondrion, and apicoplast matched that found in classical works[9,10,29,30] and allowed us to confidently assign and reconstruct these organelles in 3D.

To compare our gametocyte data with the more extensively studied ABS[17], we reconstructed representative cells for different stages in the asexual replication cycle (Fig. 1C, D, Fig. S4, Movie S3). Overall 3D morphology and appearance of the structures in individual slices match those observed in the schizont preparations by Rudlaff et al.[17] (publicly available from EMPIAR-10392). The ring and trophozoite stages have an unbranched apicoplast and mitochondrion and a single nucleus that conforms to the space constraints of the stage (Fig. S4). In contrast, the early schizont contains ten distinct nuclei and extensive mitochondrial and apicoplast networks that permeate the whole cell. The late schizont still contains an extensive mitochondrial network but has already fully divided apicoplasts and a clear organization with the nuclei more towards the outside of the parasite each with an associated apicoplast and a pair of rhoptries. The segmented schizont is comprised of 31 fully formed daughter merozoites, each containing mitochondrion, apicoplast, rhoptry pair, ER, micronemes, and nucleus, and an additional (32nd) merozoite that lacks a nucleus. Even at this late stage of development, without exception, each merozoite is still connected to the residual body that surrounds the food vacuole (Fig. 1C, Movie S3). This connectivity is highlighted by one merozoite that is connected to the residual body through an elongated neck/tube (Fig. S5A). Furthermore, the residual body is connected to the RBC cytoplasm via two continuous tubes, presumably facilitating continued sustenance to the daughter merozoites until rupture. Our data also contain occasional extracellular merozoites, most likely due to schizonts rupturing in the interval between magnetic separation and fixation. Comparing intraerythrocytic to the released extracellular merozoites, we find that volumes of the overall merozoite and individual organelles are very similar, supporting the notion that the corresponding segmented schizont is close to full maturity (Fig. 1D). The only dimension on which the extracellular and intracellular merozoites differ is that the extracellular merozoite is spherical while the intracellular merozoites conform to a tooth-like shape, likely due to space constraints in the schizont and to maintain a connection to the residual body throughout development. This observation is consistent with data from *Plasmodium knowlesi* and a post PVM-rupture *P. falciparum* schizont[17,31]. Other characteristics of the ABS are described in comparison with the gametocyte in other sections below.

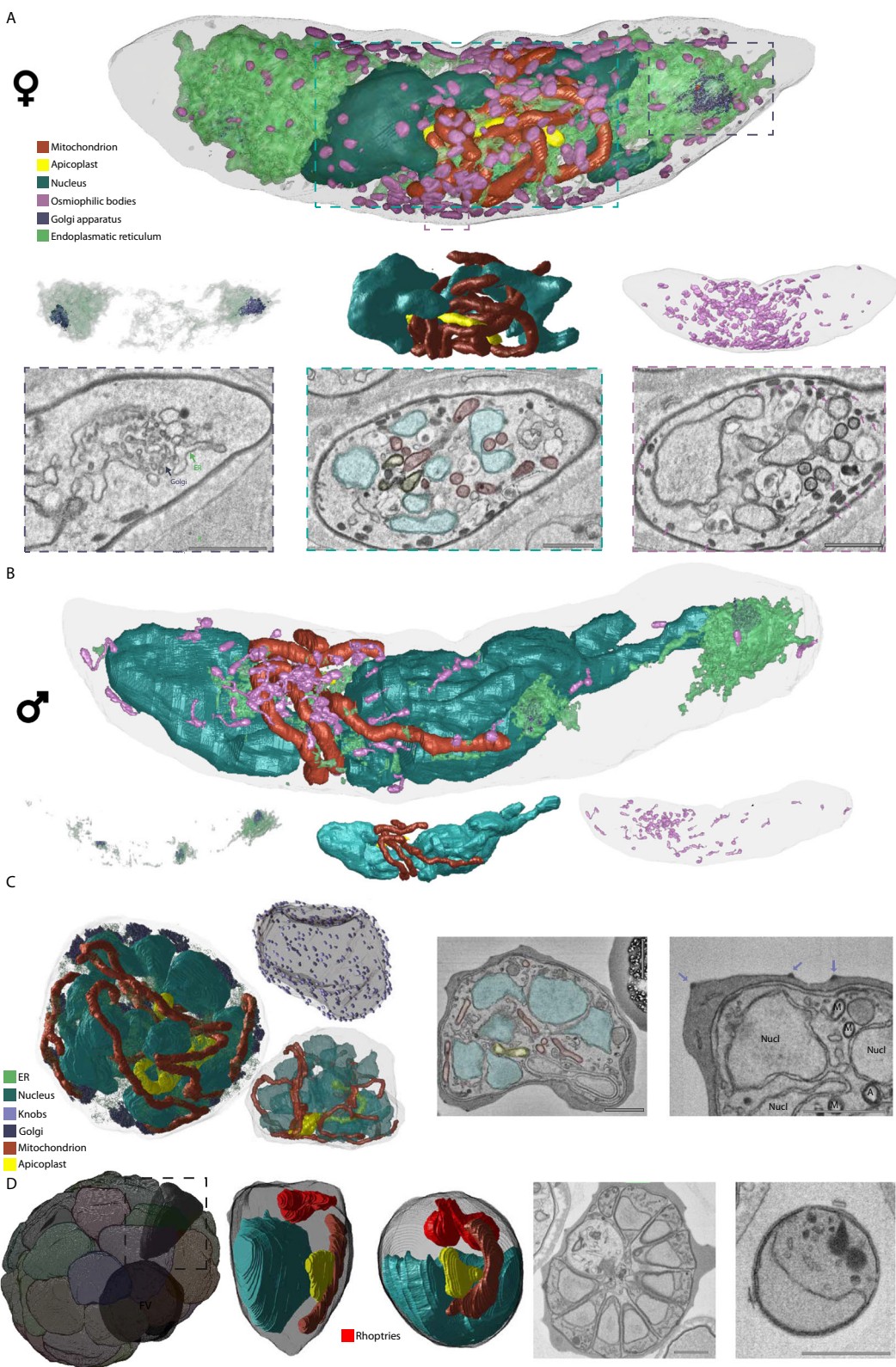

## Osmiophilic bodies show sexual dimorphism

Osmiophilic bodies (OBs), named due to their strong affinity to osmium tetroxide staining, are vesicles that are abundant in females and play a role during gametocyte egress from the RBC. *P. falciparum* males have been suggested to contain fewer OBs or even lack them completely[9,32]. Our data show that *P. falciparum* males contain vesicles that share morphological characteristics with OBs (Fig. 1B, Fig. 2I–K, Movie S2). As in *Plasmodium berghei*, these vesicles were fewer in number in males than in females[33]. To investigate this apparent dimorphism, we applied deep learning (DL) based segmentation to the stacks of five male and five female mature gametocytes, respectively. In doing so, we were able to generate renderings and extract size and shape parameters of the OBs in each cell (Fig. 2). We confirmed that female gametocytes contain both more and bigger OBs, resulting in on average ~8x more total OB volume (Fig. 2F–H). Furthermore, both aspect ratio and sphericity measurements suggest that the vesicles

**Fig. 1 | Ultrastructural features and renderings of mature gametocytes, schizont and segmented schizont. A** Rendering of ultrastructural features of a mature female gametocyte and specific renderings of Golgi and ER distribution (left panel), relation of nucleus/apicoplast/mitochondrion (middle panel) as well as distribution and appearance of osmiophilic bodies. Appearance of rendered structures in exemplar micrographs is matched to rendered features through color-coded dashed lines. Scale bars = 1 μm. **B** Rendering of ultrastructural features of a mature male gametocyte and specific renderings as in (**A**). Note the relative paucity of ER, Golgi, and osmiophilic bodies relative to the female gametocyte. **C** Rendering of ultrastructural features of an early schizont and specific rendering of relation of

nucleus/apicoplast/mitochondrion as well as individual knob distribution. Appearance of ultrastructural features is shown in two exemplar micrographs. Scale bars = 1 μm. M Mitochondrion, ER Endoplasmic reticulum, Nucl Nucleus, FV Food vacuole. **D** Rendering of daughter merozoites within a segmented schizont and rendering of one intracellular and one extracellular merozoite with ultrastructural features. Overview on appearance of the two merozoite shapes is shown in two exemplar micrographs. Scale bars = 1 μm. For gametocytes all observations are based on and consistent throughout four biological replicates. The asexual blood stage observations are based on two biological replicates.

---

observed in male gametocytes are moderately more spherical and less elongated (Fig. 2D, E). The renderings of the dataset further suggest that OBs in females appear more prevalent away from the polar ends of the cell and are often found in dense clusters (Fig. 2I, K), while in males distribution appears more even throughout the cell (Fig. 2K). Moreover, in both sexes we frequently identified tail-like extensions of the OBs, which appeared more pronounced and elongated in male gametocytes. As the DL-based segmentation model was unable to accurately segment out the tail-like extension, their relative prevalence was counted manually in one representative male and one representative female mature gametocyte. In doing so, we arrived at 125 total OBs of which 110 (88%) were tailed for the male gametocyte and 331 OBs of which 245 (74%) were tailed for the female gametocyte. While this tail has been observed previously in a thin section of a female *Plasmodium cathemerium* gametocyte and a *Plasmodium gallinaceum* gametocyte[30], and has been described in text in a previous EM investigation of *P. falciparum* gametocytes[10], it appears largely absent from previous microscopic data of *P. falciparum* or *P. berghei* gametocytes. These infrequent observations may reflect actual differences in underlying biology of the analysed parasites. More likely though they stem from methodological differences in gametocyte culture conditions, preparations, and staining or the use of high-resolution volumetric data as opposed to individual thin slices. In the rodent malaria parasite, *P. berghei*, male osmiophilic bodies have been shown to have a distinct but overlapping proteome compared to OBs identified in female gametocytes[33], which could underpin the subtle differences in OB morphology we observe between sexes. We hypothesize that the putative OBs identified in male *P. falciparum* gametocytes also differ in protein composition as males do not stain positive for the Pfg377 protein that readily stains OBs of female gametocytes[32]. Supporting this hypothesis a recent preprint suggests presence of two functionally distinct subtypes of vesicles with different proteomes that aid in gametocyte egress, only one of which – in line with our data - shows any evidence for sex specificity[34].

### Parasite-induced modifications of the red blood cell

Both asexual and sexual blood-stage malaria parasites extensively remodel their host RBC inducing extraparasitic membranous structures (Fig. 3). In ABS-infected RBCs, we observed the characteristic membranous stacks called Maurer's clefts (Fig. 3C). In both male and female gametocytes, extraparasitic structures were less numerous, unevenly distributed, and generally larger though heterogeneous in size (Fig. 3A, B). RBCs infected with mature gametocytes contained distinct flat, membranous disks. Furthermore, we consistently find a higher density of extraparasitic structures in the Laveran's bib that spans between the two ends of the mature gametocyte[35]. This matches results of previous work that found accumulation of a class of exported proteins to this subcellular location[36]. Finally, we find an occurrence of a 'Garnham body', a rare gametocyte exclusive structure originally discovered in 1933 with still unknown functional significance[37] (Fig. S6). Our data confirm the highly membranous appearance of the Garnham body described in previous micrographs and specifically we identify four double membranes. However, we

found no associated haemozoin, which was identified in some light microscopy examples[9,10,29,38] (Fig. S6A, B). The Garnham body was found adjacent to but not within the Laveran's bib, at a site devoid of IMC and situated in the vicinity of the opening of the cytostome (Fig. S6C). The presence of a Garnham body in the RBC also coincides with electron dense protrusions of the parasite vacuolar membrane radiating in all directions from the parasite and a membranous electron lucent compartment within the parasite, that were observed in no other cells (Fig. S6A, B). These concurrent exclusive features could indicate Garnham bodies to be a sign of an unhealthy gametocyte and/ or RBC, or suggestive of a rare subtype in gametocyte populations.

The overall surface profile of gametocyte-infected RBCs appeared to be smoother than of those infected with schizonts, of which the RBC infected with a segmented schizont demonstrated the roughest surface with considerably more peaks and valleys (Fig. S5B–D). Consistent with published literature, the plasma membrane of gametocyte-infected RBCs is not visibly modified, while the ABS-infected cells are thoroughly covered by uniformly distributed knobs, which are proteinaceous parasite-derived protrusions that mediate cytoadherence of ABS-infected RBCs[39] (Fig. 1C, Movie S3). Reconstructions suggest a total of 390 knobs on the surface of the schizont-infected RBC corresponding to 2.9 knobs/μm², which is in line with previous measurements obtained for NF54-infected RBCs via atomic force microscopy[40]. The young trophozoite is the earliest stage at which we see extensive extraparasitic structures in the RBC cytosol with smaller and less numerous knobs already appearing on the surface of the RBC (Fig. S4). As anticipated, we were able to distinguish the trilaminar membrane architecture of the gametocyte-RBC interface, consisting of the parasite vacuolar membrane (PVM), the parasite plasma membrane (PM) and the IMC. Localized areas of further thickened IMC membrane might represent the leading edge of IMC plates[20] while areas without IMC cover in immature stages suggest ongoing IMC plate biogenesis (Fig. S7A, B). In mature stages, we occasionally find IMC overhangs of unknown significance (Fig. S7C).

### The gametocyte cytostome

RBC-parasite interactions are not restricted to host-cell modification, but also include the internalization and digestion of host-cell cytoplasm to fuel parasite growth. An unusual feeder organelle, the cytostome (meaning cell mouth), facilitates this uptake of RBC cytoplasm in ABS[16]. In our micrographs of ABS and other studies[41], the cytostome presents itself as a haemoglobin filled tube surrounded by a double-membrane and possessing an electron dense "neck", called the cytostomal collar, at the invagination site (Fig. S8). Gametocytes similarly rely on haemoglobin internalization and digestion. Putative morphologically diverse cytostomes have been assigned in past EM studies[9,10]. When trying to define the cytostome in mature gametocytes, we came across likely yet distinct candidate structures. Presenting a clear invagination of the parasite membranes, the structures are always surrounded by extensive ER but otherwise can be subdivided into three categories (Fig. 4A, B). In 44% of the evaluated examples ($n = 50$), the organelle has a membrane delimited electron dense ring that is 100–160 nm thick and connected to the RBC

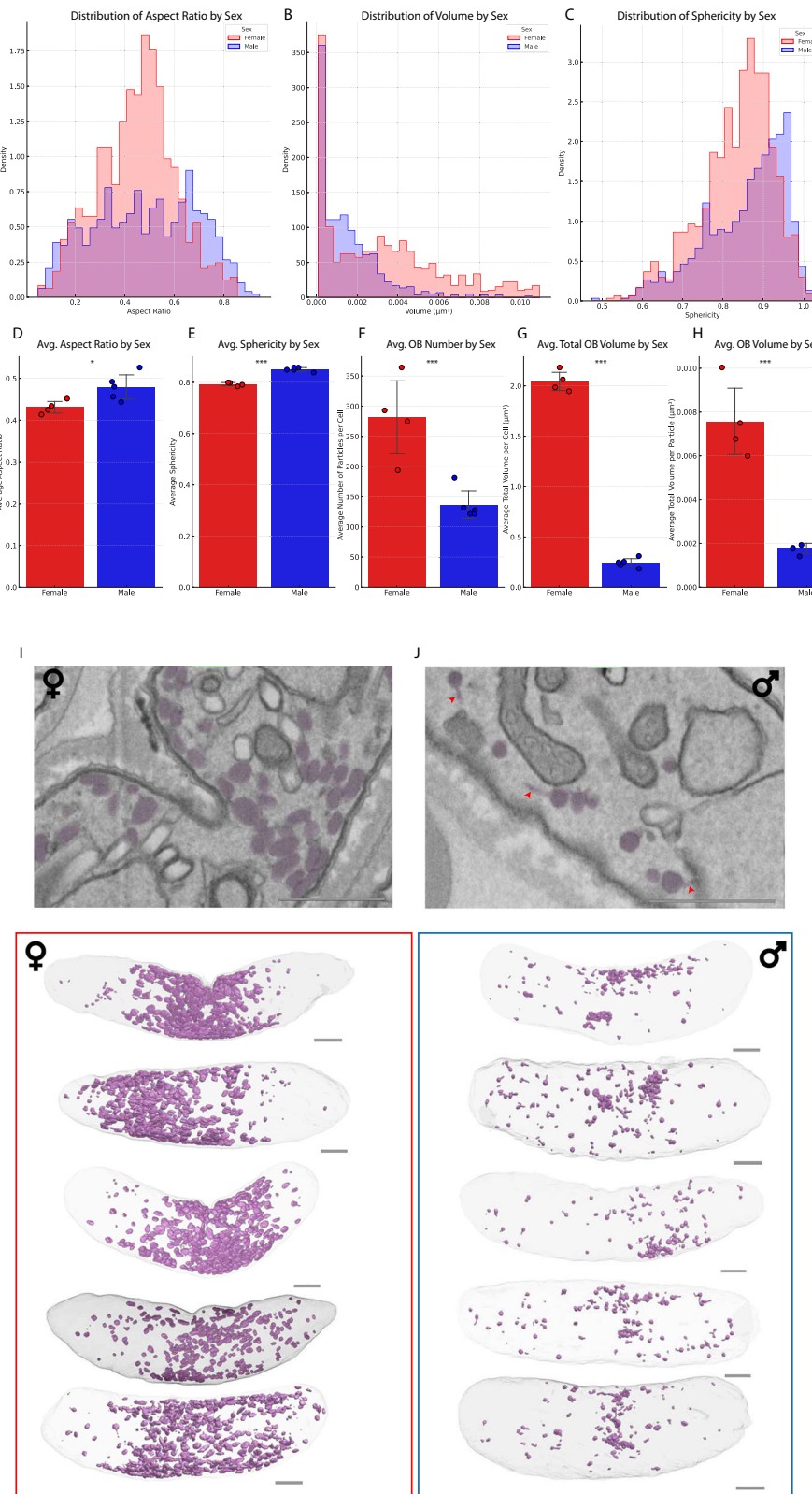

**Fig. 2 | Morphological characteristics of osmiophilic bodies in mature gametocytes. A**–**C** Histogram of aspect ratio (**A**), volume (**B**) and sphericity (perfect sphere = 1) (**C**) of osmiophilic bodies (OBs) subdivided by sex. **D**–**H** Comparison of OB morphology in male (blue) and female gametocytes (red) *N* = 10. Significance of measured differences was tested with a two tailed *t*-test and significance was denoted with * (*p* < 0.05), **(*p* < 0.01) or *** (*p* < 0.001). SEM is denoted with vertical bars. Exact *P*-values for the different parameters are: Aspect ratio = 0.03, Sphericity = 0.00001, Particle number = 0.0034, Total OB volume = 0.0001, Average OB volume = 0.0001. I, **J** Representative micrographs of a female (**I**) and male (**J**) gametocyte showing differing density and shape of osmiophilic bodies (purple hue) in the respective sexes. Tail-like extensions of OBs in the male micrograph are indicated with red arrows. **K** Comparative rendering of OBs (purple) in all underlying female (red border) and male (blue border) gametocytes used for above analysis. Scale bars = 1 μm.

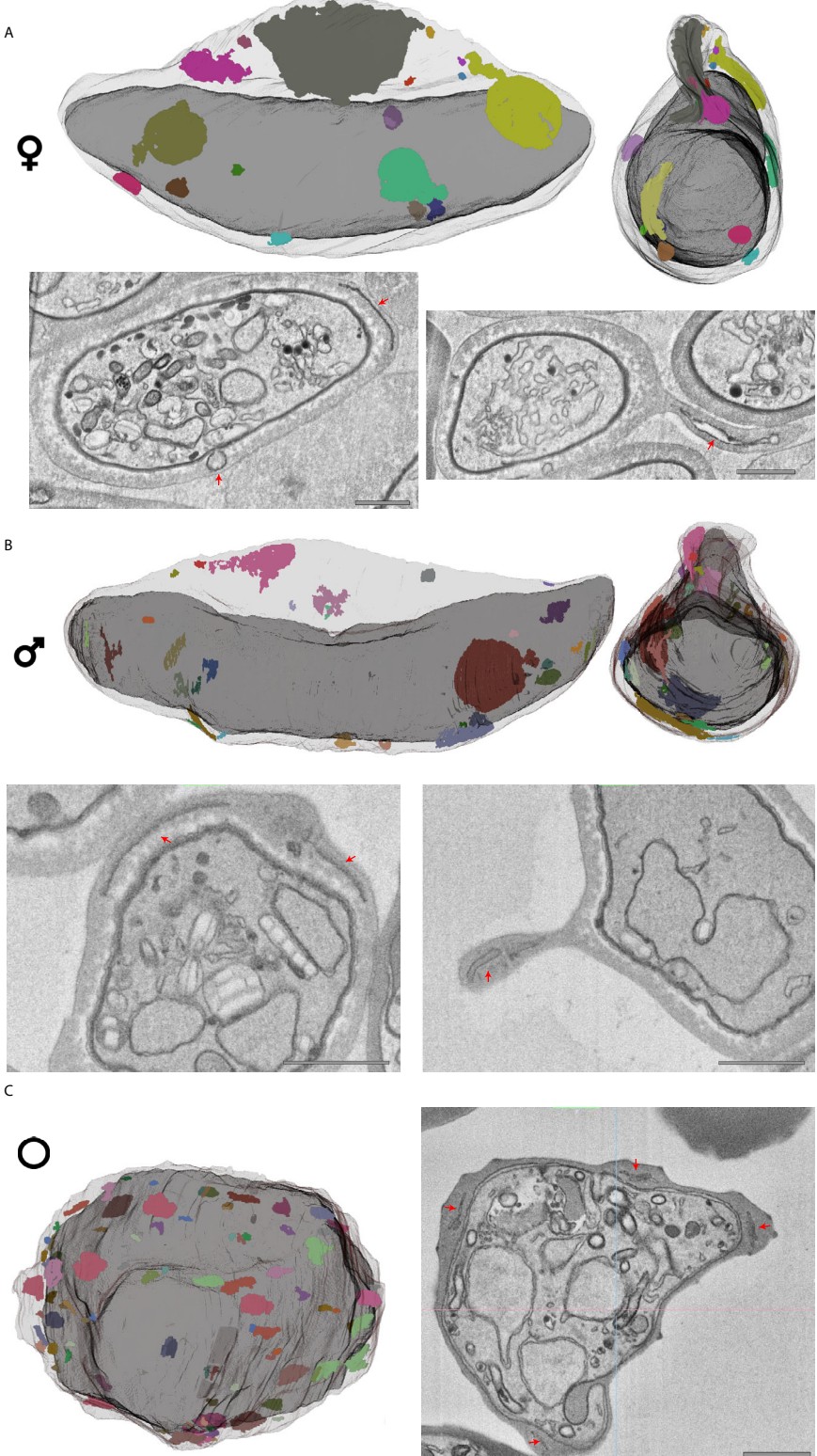

**Fig. 3 | Extraparasitic structures differ in gametocytes and ABS. A** Mature female and (**B**) male gametocyte, and (**C**) schizont rendered with RBC outline and extraparasitic structures in various arbitrary colours to distinguish separate and connected extraparasitic structures. Gametocyte renderings are shown from two angles. Examples of extraparasitic structures in micrographs are highlighted with red arrows. Scale bars = 1 μm. For gametocytes all observations are based on and consistent throughout four biological replicates.

cytoplasm while the lumen circumscribed by the ring appears electron lucent and can occasionally contain further membranous or dark stained structures. In 40% of the cases, the structures are similar to those described in ABS parasites with no clear delineation into ring and lumen. Circular grooves on the inside of the membrane that are absent in the ABS cytostome may be indicative of future emergence of the ring structure. In the remainder the ring is relatively larger with inconsistent thickness and a smaller electron lucent internal lumen

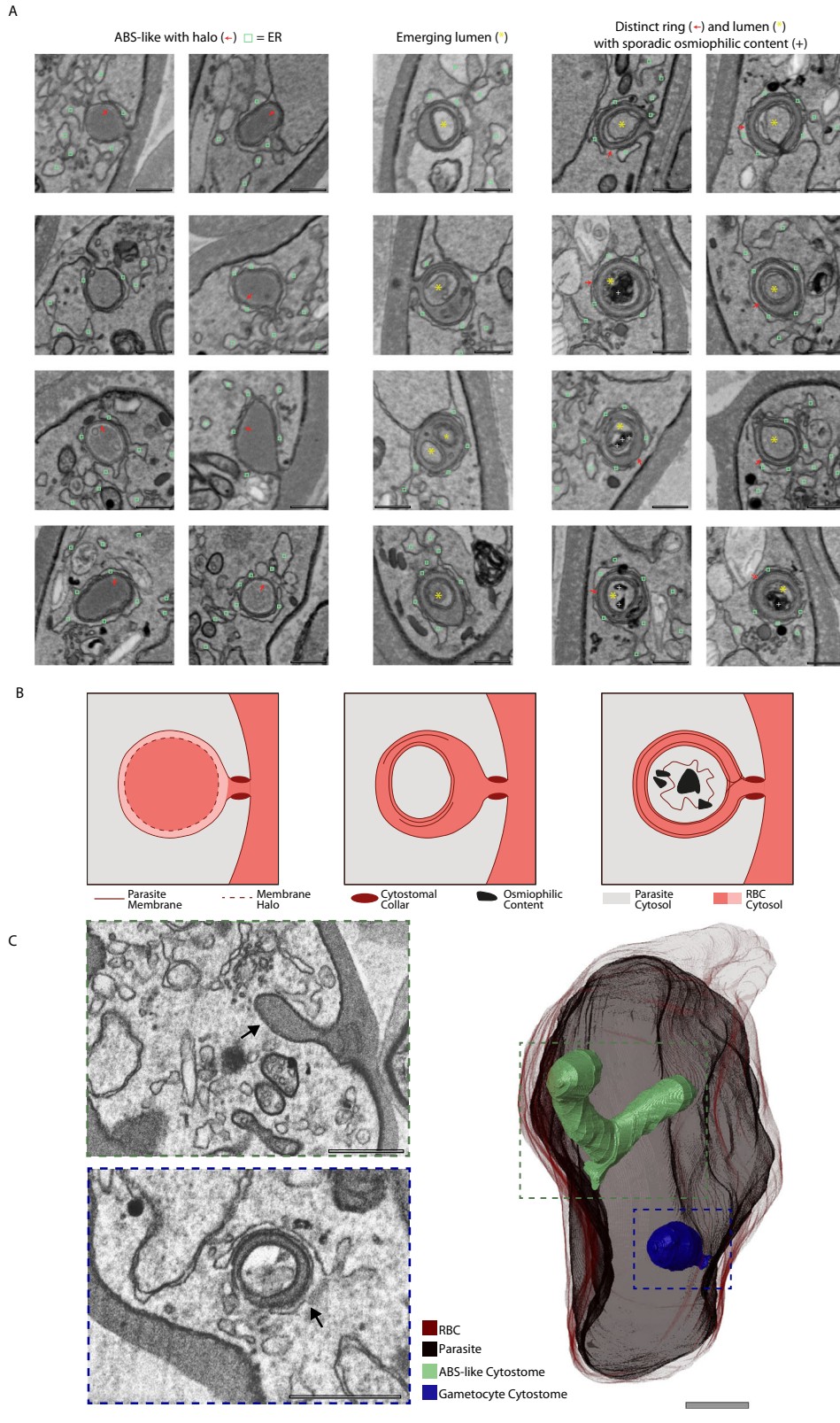

potentially representing an in-between state. Furthermore, it appears that each gametocyte that we have fully imaged only contains a single cytostome of any of these three gametocyte-specific subtypes. Conversely, in ABS multiple cytostomes have been reported[42] and are also evident from our data with a maximum of seven cytostomes per cell (Fig. S8, panel 3). Interestingly, in some developing gametocytes, we find both a canonical ABS-type cytostome and a gametocyte variant (Fig. 4C). This observation suggests that the structure we identified in

gametocytes might be entirely distinct from the cytostome. In mature stages, haemoglobin internalization is largely finished and consequently the cytostome may no longer be required for its canonical purpose. It is tempting to speculate that this cytostome variant is instead involved in host lipid acquisition, as parasite lipid content dramatically increases throughout gametocytogenesis[43]. This would also offer a plausible explanation for the ER, as the primary site of lipid metabolism, consistently surrounding the cytostome and would

**Fig. 4 | The gametocyte cytostome. A** Exemplar micrographs of the cytostome in 20 different gametocytes subdivided into three categories. Left: Cytostome without further membrane delineation in lumen, relatively homogenous lumen and circular grooves that could indicate future emergence of ring. Middle: Relatively large membrane electron dense ring and small electron lucent lumen. Ring is continuous with RBC cytosol, putative intermediate state. Right: Electron dense ring with homogenous thickness and lumen with heterogeneous content. Ring is continuous with RBC cytosol. All categories have close association of ER unlike the ABS cytostome. ER = light green square, halo/membrane ring = red arrow, yellow asterisk = separate lumen, plus sign = osmiophilic content, scale bars = 0.1 μm.

**B** Schematic of the hypothesized cytostome variants/intermediates. Each schematic corresponds to the variant above. **C** Rendering of a developing gametocyte that contains both a canonical ABS cytostome and a cytostome with distinct ring and lumen. The ABS cytostome (green) is characterized by homogenous dark lumen and continuation of the parasite membrane that forms the invagination. The gametocyte cytostome (blue) is characterized by a bulbous shape and distinct electron dense ring with electron lucent lumen. Coexistence of ABS-like and gametocyte cytostome appears uncommon and has been observed in three cells throughout all stacks. Scale bars = 0.5 μm.

suggest that the occasional dark contents of the lumen seen in our preparations are (osmiophilic) lipids. It would be informative to test whether the usually drug-resilient late-stage gametocytes are particularly sensitive to inhibitors of stage-dependent lipid metabolism and whether known gametocidal activity of inhibitors of lipid metabolism such GT11 and MSDH-C[44] (that are ineffective against ABS) are reflected in perturbed morphology of the gametocyte cytostome. To further delineate whether ABS and gametocyte cytostome are distinct structures, inducible mislocalizations or immunoelectron microscopy of Kelch13 as demonstrated by Tutor et al.[45] but applied in the gametocyte stages could provide clarity. These would be particularly interesting as K13 mutations are the main driver of resistance against artemisinin, the standard first line treatment against malaria[42].

## The gametocyte ER and Golgi-apparatus

Universally the ER is an extensive interconnected organelle continuous with the nuclear envelope that serves as a transport hub, hosts various critical cellular functions, is widely connected with other organelles, and even acts as a mediator of organellar interactions[46]. We find that the ER in both immature gametocytes and mature female gametocytes is extensive and occupies large parts of the cell, similar to the schizont (Fig. 1A, Fig. S2, Movie S1). The mature male gametocyte on the other hand appears to contain much less ER, which is in line with previous observations and proteomic data[47,48] (Fig. 1B, Movie S2). It is noteworthy that in the mature female gametocyte, the ER is most dense in the polar regions of the parasite. This coincides with the distribution of a closely associated organelle, the Golgi, which, in *Plasmodium*, takes the rudimentary form of dispersed unstacked cisternae and is recognizable as a smooth membraned vesicle cluster[29,49–51]. In females, the Golgi is predominantly found in two distinct clusters at both polar ends, while in the mature male it appears distributed across smaller clusters, similar to the developing gametocytes that also have a more widespread distribution. Compared to the gametocytes, the Golgi is much more extensive in the schizont and found in evenly dispersed clusters around the cell (Fig. 1C, Movie S3).

## The ER as a putative path across the IMC

The ER also facilitates the extensive host-cell remodelling that occurs in ABS. In *P. falciparum*, an estimated 300+ unique proteins, >5% of the proteome, is exported to the host cell to modify aspects such as cytoadhesion, the host cytoskeleton, or nutrient permeability[52]. Proteins destined for export are processed in the ER and delivered to the PM via vesicular transport and then forwarded to the *Plasmodium* Translocon of EXported proteins (PTEX) in the PVM, which translocates the proteins into the host cell. While extensively researched in ABS, we know relatively little about how this process translates to the gametocyte situation. Maurer's clefts appear absent in gametocytes, possibly replaced by other cleft-like structures (Fig. 3A, B)[53], and the gametocyte exportome is distinct from the ABS exportome[54]. Whilst protein export machinery is essential for early gametocyte development[55,56], protein export has not been conclusively demonstrated in gametocytes beyond stage III. Indeed, as the IMC cover increases, staining of the export machinery on the PVM becomes less pronounced or even disappears completely in *P. falciparum* and

*P. berghei* gametocytes, respectively[54,57]. If protein export still happens in late gametocyte stages the mechanism by which the additional barrier presented by the IMC is overcome, is currently unknown[58]. In mature gametocytes, we always find sites at which the ER runs very close to or is in direct contact with the parasite membrane (Fig. S9). From our observations this interaction can manifest in a few different forms. The most frequent observation is that ER directly contacts the IMC, leading to a continuum between these two compartments as well as local disruption of the IMC (Fig. S9A). Less frequently but consistently, we observe "budding" of a piece of ER that contacts the parasite membrane (Fig. S9B). These local continuities between ER and IMC might represent a mechanism to bridge the IMC and allow canonical ER-derived vesicle fusion with the PM or facilitate alternative means of protein export. In developing gametocytes with incomplete IMC cover, we find that all extension sites involve extensive ER interaction with the nascent IMC (Fig. S9D). This is in line with previous EM-based observations that the outer nuclear envelope interacts with nascent IMC and the PPM in developing gametocytes[19] and with immunofluorescence microscopy data suggesting regions of ER contact with the IMC[59]. Even in mature gametocytes with complete IMC cover we find sites at which ER is continuous with electron dense material that resembles IMC (Fig. S9C). Taken together, these observations, while anecdotal, provide further evidence for a dynamic interplay between ER and IMC biogenesis or maintenance and suggest a speculative mechanism to bridge the additional membrane layer in gametocytes for protein export.

## Features of *Plasmodium* mitochondria

In gametocytes, the mitochondria are readily recognizable based on their cristate appearance and two-layered membrane. The cristae appear to have some degree of interconnectivity, consistent with previous interpretations of tubular cristae, but the small size of the structures and variability between slices, even at a z-resolution of 15 nm, makes clear assertions of connectivity challenging (Fig. S10B, Movies S4, 5). While boundaries of the individual cristae are more evident from the lower noise serial sectioning data, the low z-resolution does not allow for confident reconstruction of their morphology (Fig. S10A). In the individual micrographs, the cristae are similar to the recently described bulbous cristae in the related apicomplexan parasite *Toxoplasma gondii*[60]. In line with previous reports, the mitochondrion is double-membraned yet acristate in ABS. In both ABS and gametocyte mitochondria, we regularly identify electron dense mitochondrial granules (EDMGs) that appear in clusters of 1-10 EDMGs and are heterogeneous in volume with a notably smaller size, propensity to be more separate and more slender shaped in ABS (Fig. 5A)[25,61]. EDMGs are distinct from the cristae observed in gametocytes based on their homogenous electron dense appearance without clear membrane delimitation as opposed to the cristae, which are formed by a membrane that surrounds an electron lucent lumen. While EDMGs have not been described before, they are readily recognizable in the image stacks of schizonts made public by Rudlaff et al.[17] indicating they are not a unique feature of our preparations. The electron dense appearance could suggest that these granules are subcompartments that are either very proteinaceous or contain

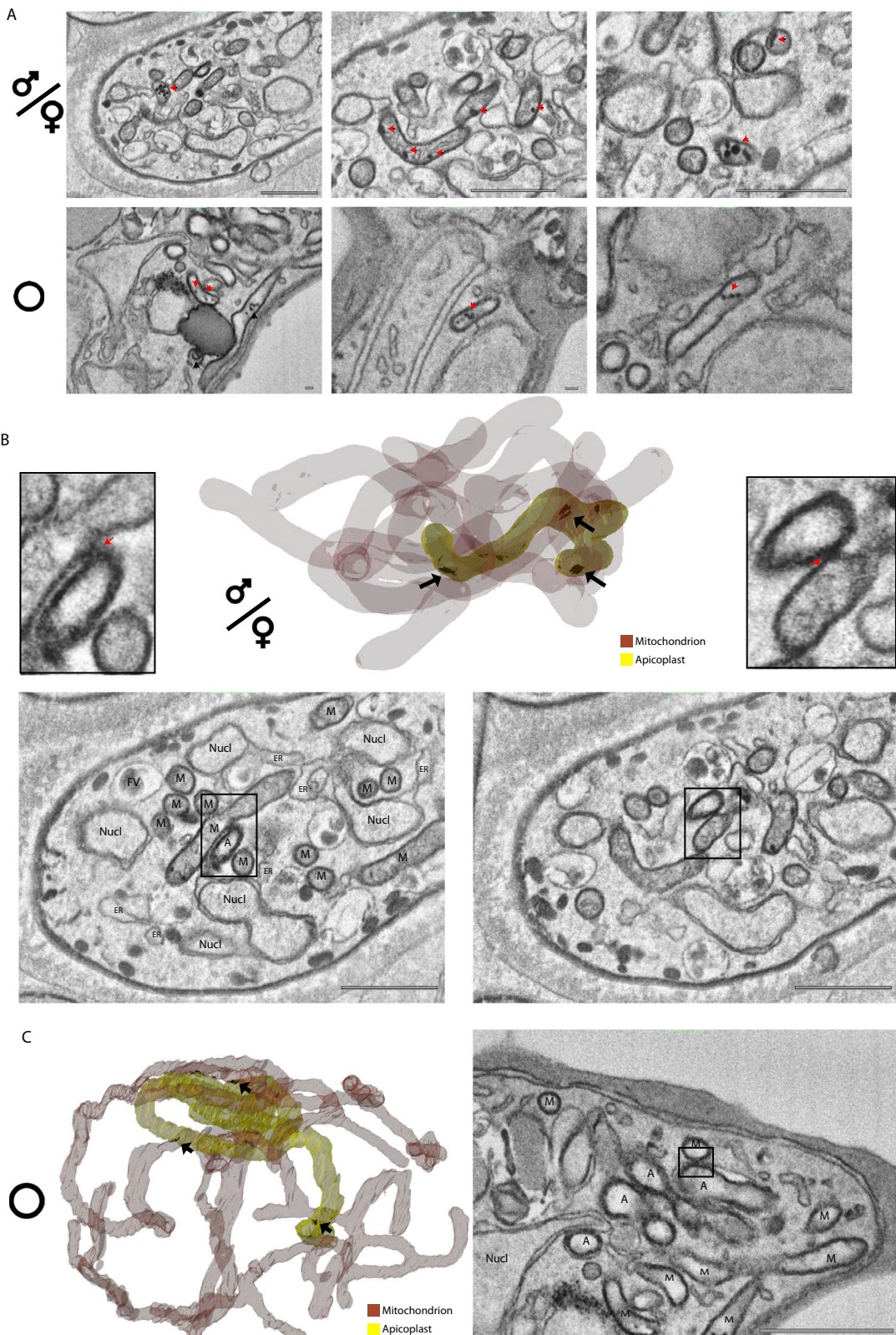

**Fig. 5 | Mitochondrial vesicles and interaction with the apicoplast.**
**A** Representative micrographs of EDMVs in gametocytes (upper panel, scale bars = 1 μm) and asexual blood stages (lower panel, scale bars = 0.1 μm). Arrowheads highlight examples of EDMVs. **B**. Rendering of exemplar gametocyte mitochondrion (red, high transparency) and apicoplast (yellow, low transparency) with putative organelle interfaces rendered in black. Representative micrographs for two interfaces and respective zoomed crops are shown below. Scale bars = 1 μm. **C** Same representation as (**B**) for a schizont. M Mitochondrion, ER Endoplasmic reticulum, Nucl Nucleus, FV Food vacuole. For gametocytes all observations are based on and consistent throughout four biological replicates. The asexual blood stage observations are based on two biological replicates. Scale bar = 1 μm.

polysaccharides, lipids, or mitochondrially relevant metals such zinc, calcium, copper, and/or iron. *P. falciparum* mitochondria have been shown to play a role in calcium mobilization and storage[62] and so-called matrix granules that are heavily enriched in calcium phosphate have been found in mammalian mitochondria[63]. In parasites isolated from fish digestive tracts, similar mitochondrial granules have been shown to store glycogen, which our staining procedure is particularly well suited to show[64,65]. Increase of energy storing granules in gametocyte stages could be a plausible preadaptation for survival in the relatively nutrient deprived environment of the mosquito midgut. Alternatively, the EDMGs could represent the mitochondrial RNA granules described in mammalian mitochondria. These are highly proteinaceous, non-membrane delimited sub-compartments in the mitochondrial matrix that are thought to play an important role in mitochondrial RNA processing, mitoribosome assembly, and gene expression[66]. As mitochondrial translation in *Plasmodium* purely relates to complex III and complex IV components of the respiratory chain, this could explain the discrepancy in EDMG size between the comparatively respiratory chain-poor ABS compared to the respiratory chain-rich gametocytes[25,67].

## Multiple mitochondria in disguise

The current consensus is that, like ABS, gametocytes contain a single heavily branched mitochondrion with diverse morphology[24]. Hence, we were surprised to find not a single but multiple mitochondria in gametocytes of all developmental stages and both sexes that, while clustered tightly with close membrane appositions, did not share a continuous lumen (Fig. 6A, B, Movies S5, 6). The number of mitochondria did not markedly differ between mature male and female gametocytes but the average volume of the individual mitochondria is larger in female gametocytes (Fig. 6C). Consequently, both the total mitochondrial volume and the fraction of the parasite that is occupied by the mitochondrion differ significantly between the sexes though they are remarkably consistent within the single sexes (Fig. 6C). Examining the mitochondrial morphology, we find branched mitochondria in both populations but while female gametocyte mitochondria have relatively consistent tubule diameter (~ 0.30 μm), male mitochondria tend to have a more varied diameter with thinner and thicker areas (Fig. S11A). To investigate this phenomenon with another imaging modality, we carefully revisited mitochondrial morphology of a larger number of mature gametocytes using fluorescence microscopy. As expected, most images gave the impression of a fully interconnected organelle, but we also found examples where at least some mitochondrial staining appeared separate from the major cluster (Fig. 6D, Movies S7–10). This is consistent with the FIB-SEM data, where completely separated mitochondria can be observed (Fig. S11B), though in most cases the mitochondria are in close proximity, potentially even suggestive of homotypic membrane interactions that are commonplace among fungal and mammalian mitochondria[68] (Fig. 6B). These short distances of < 50 nm are well below the diffraction limit of conventional fluorescence microscopy and would appear as a continuous organelle. We observed no morphological indications for poor gametocyte health, which was confirmed by normal gamete activation and the ability of male exflagellation. Moreover, exflagellating males have clearly distinct, dispersed, and rounded mitochondria (Fig. 6D, Movies S11, 12). In our FIB-SEM data, we identified one comparable example of a male gametocyte showing bloated, rounded, and dispersed mitochondria that appeared to be in the process of separating, though at this point it is unclear whether this is a male gametocyte at the onset of activation or an aberrant cell (Fig. S11C). We find occasional mature male gametocytes showing a similar pattern using fluorescence microscopy (Fig. S11D). Generally, cristae appeared homogenous across the multiple mitochondria within a gametocyte and were also observed in the bloated and dispersed phenotype (Fig. S11C, E). In comparison, the early schizont possesses a single mitochondrion with a continuous lumen, consistent with previous observations, suggesting that the observed phenotype is not directly caused by fixation or sample processing (Fig. 1C)[17]. Taken together, we find evidence for deviation from the current consensus of a fully interconnected mitochondrion in gametocytes but not ABS as well differing mitochondrial morphology between mature male and female gametocytes. The functional significance of these deviations and appearance in other strains and culturing systems remain to be investigated.

One possible function of varying mitochondrial numbers might be that it affords higher flexibility to adapt to the more metabolically varied environments that the gametocyte encounters relative to ABS. While the mitochondrial numbers we observe are within a relatively narrow range, it would be interesting to see whether varying glucose, oxygen or amino acid concentration in the medium would lead to shifts in the mean number or volume of mitochondria. Another tempting explanation is that the highly increased OXPHOS content of gametocytes[25] leads to higher demand for translation products of the mitochondrial genome as the only three genes encoded on the mtDNA are subunits of OXPHOS complexes. Rather than relying on diffusion or transport of transcripts or translation products across the whole organelle, it might be more efficient to have multiple smaller mitochondria. In line with this hypothesis, it was found that gametocytes contain seven times more mitochondrial DNA than ABS[69]. As this estimation is close to our observed mitochondrial number, it is tempting to speculate that mtDNA content per mitochondrion between ABS and gametocytes remain constant but that the number of mitochondria drives this difference. Gametocytes are also the stage where malaria parasite mitochondria form cristae de novo. As both mitochondrial fission and cristae formation have some overlap in canonically required molecular machinery[70,71] and are dependent on similar lipid moieties[72,73], one could envision some coordination and interdependence of these processes. Finally, the primary site of reactive oxygen species (ROS) production in eukaryotic cells is the mitochondrion with a higher respiratory activity being mechanistically linked to higher rates of ROS production[74,75]. Gametocytes are much longer lived and rely much more on aerobic respiration than ABS[25,76]. Consequently, they are at risk of accruing more oxidative damage and may have developed strategies to better manage the production of or harmful effects by ROS. Mitophagy is the major eukaryotic mitochondrial quality control mechanism to remove damaged mitochondria but such a strategy is only feasible if a cell harbours multiple mitochondria[77]. It would be informative to test whether mitochondrial turnover takes place via e.g. pulse labelling experiments or investigate the presence and relevance of homologs to mitophagy-related proteins that have been identified in previous work[78]. A strong indicator for the relevance of organellar cycling is that a set of antimalarials appears to derive their effect from disrupting the interaction between the apicoplast-resident Atg8 and the autophagosome protein Atg3[79]. With the mitochondrion and the parasite's oxidative state being validated antimalarial drug targets, exploring these various aspects and processes of gametocyte mitochondrial biology could help develop new (synergistic) gametocidal agents.

## Appositions of mitochondrion and apicoplast

In gametocytes, the mitochondria form a network that is wrapped around the apicoplast (Fig. 1A, B, Movies S1, 2). The apicoplast is recognizable as a clearly distinct structure from the mitochondrion due to the thicker appearance of its four surrounding membranes and lack of internal membranous structures and more electron lucent lumen. The mitochondrion and apicoplast are not spread throughout the whole cell but localized to a relatively central area of the gametocyte, always adjacent to the nucleus. It is noteworthy that this association and the mitochondrial network are much tighter than what we observe in schizonts or what has previously been described for *P. berghei* liver-

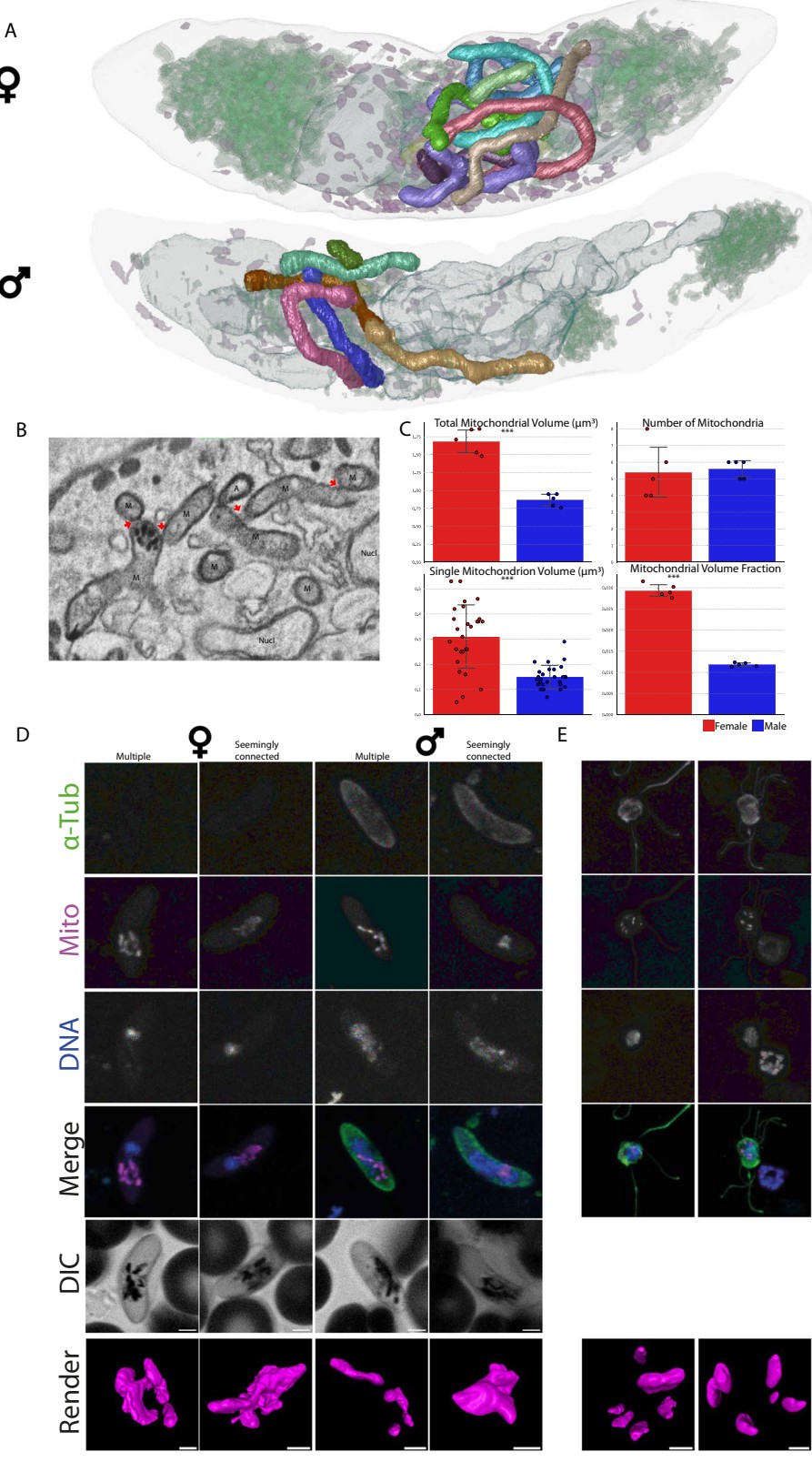

stage parasites[80] where the mitochondrion permeates the whole cell (Fig. 1C, Movie S3). Yet, the tight association is clearly distinct from the spatial separation of these organelles in sporozoites[81]. Next to a general vicinity of the two organelles, we observe electron dense junctions spanning the membranes of the two organelles in both gametocyte and ABS (Fig. 5B, C). While it remains unresolved whether these sites constitute true membrane contact sites (Note S1), the thickened electron dense interaction area is indicative of a tethering structure, providing a reasonable explanation why the organelles are consistently co-purified[82]. While a putative physical connection does not provide mechanistic evidence, the observed phenomenon may facilitate metabolic cooperation, *e.g.* by serving as an exchange site for metabolites of the haem biosynthesis pathway for which the enzymes are localized partly in the mitochondrion and partly in the apicoplast[83,84].

**Fig. 6 | Gametocytes contain multiple mitochondria in close vicinity.**
**A** Rendering of mature female (upper panel) and male gametocyte (lower panel). Each mitochondrion is rendered in a separate colour. The male gametocyte has six distinct mitochondria while the female gametocyte has nine distinct mitochondria. In the female gametocyte membrane appositions are more frequent and mitochondria are more closely associated. Nucleus, osmiophilic bodies, ER, Golgi, and apicoplast are rendered with high transparency to provide cellular context.
**B** Exemplar micrographs showcasing membrane apposition between different mitochondria indicated by red arrows. The mitochondrial membrane remains intact at these sites and no continuity between the different mitochondria is evident. Membrane appositions are consistent throughout gametocytes in all four biological replicates. Scale bar = 0.1 μm. M Mitochondrion, A Apicoplast, Nucl. Nucleus. **C** Bar-graphs with average number and volume metrics of mitochondria in mature male and female gametocytes. Significance of measured differences was tested with a two tailed t-test and significance was denoted with * ($p < 0.05$),

** ($p < 0.01$), or *** ($p < 0.001$), $N = 10$. SEM is denoted with vertical bars. *P*-values for the different parameters are: Total Volume = 0.000015, Number of Mitochondria = 0.805, Single mitochondrion volume = 0.0000001, Mitochondrial Volume Fraction = 0.000000008. **D** Immunofluorescence analysis of mature and female gametocytes. Depicted from left to right are maximum intensity projections using anti-α-tubulin antibodies, MitoTracker™, DAPI, a merge of all channels, and differential interference contrast (DIC) images. α-tubulin was used to distinguish male and female gametocytes, DNA was visualized using DAPI, and mitochondria were visualized using MitoTracker™. For each sex one example was given for recognizably separate mitochondria and a seemingly interconnected mitochondrion. Scale bar = 2 μm. Additional renderings of the mitochondrion were generated based on the mitochondrial fluorescence signal (bottom row, scale bar = 1 μm). **E** Immunofluorescence analysis of exflagellating male gametes. Channels are the same as in (**C**). Mitochondria are clearly separate and dispersed throughout the cell body.

## Appositions of ER and mitochondrion

From mammalian and yeast mitochondria, we know that the ER is intimately linked to and interacts with the mitochondrion through membrane contact sites (MCS). In these species, MCS cover 2–5% of the mitochondrial surface area and are composed of a known set of interactive proteins[85,86]. These sites are thought to be crucial for lipid homeostasis and calcium transport. In our data, we similarly find multiple sites where the ER is in very close proximity to the mitochondrion (Fig. S12A, B). Anecdotally, both in the schizonts and gametocytes, we find examples where these sites are accompanied by EDMGs appearing to span the mitochondrial membrane and contact the ER (Fig. S12), which provides tentative support for a role of EDMGs in calcium storage and/or mobilization. The appearance of the ER-mitochondrion appositions resembles micrographs of previously described ER-mitochondrion MCS[87], leading us to believe that they are conserved in *Plasmodium*. However, from all previously described tethering complexes, the *Plasmodium* genome is lacking at least one critical component[88–91]. In *Saccharomyces cerevisiae*, the ER membrane complex (EMC) and TOM5 facilitate phospholipid exchange between ER and mitochondrion[92]. Recently, we demonstrated that an assembled EMC is present in *P. falciparum*[25], but we could not identify a TOM5 orthologue in the *Plasmodium* genome. This prompted us to re-interrogate the underlying complexomics data and look for another potential mitochondrial interactor with the EMC. In doing so, we found that *Pf*TOM7, another component of the translocase of the outer membrane (TOM) complex, comigrates in native electrophoresis with the EMC and TOM complexes (Fig. S14A). Looking at the multiple sequence alignment, few residues are conserved between PfTOM7 and *Sc*TOM7/TOM5 (Fig. S14B, C). While overall structural features of *Pf*TOM7 and *Sc*TOM7 appear similar based on alignment of the *Pf*TOM7 AlphaFold prediction[93] and experimentally determined structure of *Sc*TOM7[94,95], *Sc*TOM5 is expectedly less similar outside of the (predicted) TM helix (Fig. S14D). This suggests that the specific binding site is unlikely to be conserved if *Pf*TOM7 indeed functions as a tether bridging the ER and mitochondrion. It might also suggest that there is biological utility in specifically making the tether through a component of the TOM complex.

## Features and dimorphism of the gametocyte nucleus

The nucleus is bounded by an inner and outer leaflet of the nuclear envelope, where the outer leaflet is continuous with the ER and the inner leaflet circumscribes the nucleus. In gametocytes, we find an intriguing deviation from the typically round or oval nuclear shape[96]. The nuclei share a bulbous body from which thinner diverse extensions extrude predominantly in one direction covering large parts of the cell (Fig. 7A, B). The bulbous body always contains a more electron dense, non-membrane-bound region, which has previously been assigned as the nucleolus of the female gametocyte[29] (Fig. 7C). The observed nuclear shapes are largely consistent with prior studies that

noted a discrepancy between a relatively small and localized DNA stain and an elongated staining from a nucleus-targeted fluorophore[19,97]. In stage IV – V gametocytes, a clear nuclear dimorphism is apparent with female gametocytes possessing a less complex smaller nucleus with an average volume of $6.6 \pm 0.3\ \mu m^3$, while males have a more complex nuclear shape and a much higher nuclear volume at $12.4 \pm 1.1\ \mu m^3$ (Fig. 7B). A comparatively enlarged nucleus in males is consistent with well-known differences observed in Giemsa-stained samples and might prepare male gametocytes for the rapid nuclear division and DNA replication that they undergo upon activation. While smaller, the female nucleus is still relatively large, possibly explained by the proposed role of the nuclear extension in elongation of the gametocyte[19]. Based on our measurements, for merozoites the same genetic material fits into a nucleus of ~0.5 $\mu m^3$ while for our exemplar schizont in the process of DNA duplication, average nuclear volume is around 1.8 $\mu m^3$. Stage II - III immature gametocytes also reflect the nuclear dimorphism observed in stage IV – V gametocytes, with some having a relatively smaller and less complex nucleus while another subset has a bigger and more complex nucleus (Fig. 7A). While we cannot confidently assign the developing gametocytes to either sex, a plausible hypothesis would be that the subset with smaller nucleus represents immature female gametocytes, while cells with larger nucleus represent immature male gametocytes. However, this is in conflict with a recent finding that commitment, as indicated by measurable differences in gene expression, to either male or female cell fate is only decided at stage III[98]. Furthermore, a recent study utilizing array tomography has found no such nuclear dimorphism in earlier stages but does show similar dimorphism and absolute volumes to our findings for later stages[19]. It would be interesting to see whether these different observations stem from differences in experimental methods, stage assignment, image segmentation or biological material.

## Appositions of nucleus and endosymbiotic organelles

In *Plasmodium*, both mitochondrion and apicoplast have essential organellar ribosomes with promising features as drug targets[99–102]. However, *Plasmodium* mitochondrial DNA does not encode the tRNAs that are required to translate its genome[103]. Recently, it has been found that the nuclear-mitochondrial MCS play a role in RNA exchange and signalling. These sites are distinct from the well-characterized peripheral ER-mitochondrial MCS but likewise enabled by tether proteins without obvious homologues in the *Plasmodium* genome[104,105]. We find distinct appositions of the nucleus and mitochondrion in both schizonts and gametocytes (Fig. S13). At these sites there appears to be continuity of the outer leaflet of the nuclear envelope and the outer mitochondrial membrane. In contrast, we find that the nucleus-apicoplast interface in ABS is characterized by a local condensation of the normally spatially separated layers of the nuclear envelope into one electron denser layer without obvious membrane fusion (Fig. S13B). Whereas the mitochondrion in

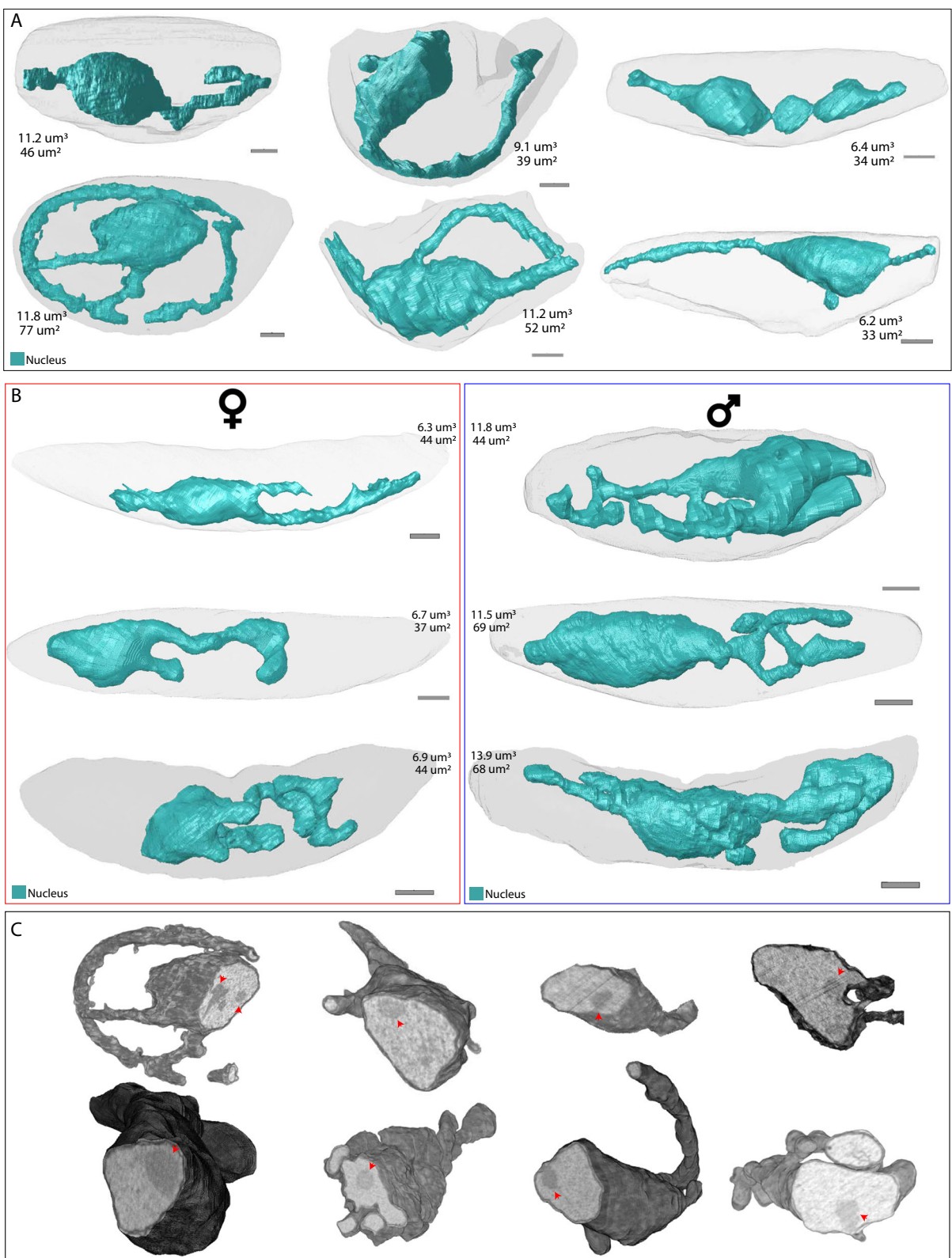

**Fig. 7 | Distinct nuclear morphology in gametocytes.** Renderings of nuclei (teal) in (**A**) stage II-III and (**B**) stage IV-V gametocytes. Stage IV-V gametocytes are divided in female (red outline) and male (blue outline) gametocytes based on the number and appearance of osmiophilic bodies, haemozoin distribution, and ER prevalence.

For all nuclei, the respective volumes and surface areas are indicated. **C** Rendering of cross section of nuclei with grey values from EM data overlaid. Red arrowheads point at the position of the putative nucleolus.

**Table 1 | Basic parameters of FIB-SEM stacks used in these studies**

|  | Sample 1 | Sample 2 | Sample 3 | Sample 4 | Sample 5 | Sample 6 |
|---|---|---|---|---|---|---|
| **Life-cycle stage** | GCT | GCT | GCT w/ ABS | ABS | ABS | GCT w/ ABS |
| **Dimensions (x,y,z) [px]** | 4268,2784,2293 | 4428,3804,952 | 4402,2850,900 | 4423,1430,1085 | 4010,3157,739 | 13044,4019,3005 |
| **Total Volume [um³]** | 10217 | 6013 | 4234 | 2573 | 3508 | 59034 |

All samples constitute biological replicates.

gametocytes closely envelops most of the apicoplast, only a subset of cells has close physical proximity between nucleus and apicoplast and in those cells we never find indications of direct membrane contact. Functionally, interfaces of apicoplast and mitochondrion with the nucleus might permit phospholipid homeostasis or RNA exchange as has been suggested in model eukaryotes[104,105]. The more intimate connections between the nucleus and the mitochondrion could provide a potential mechanism for import of tRNAs into the organelle.

### Insights and what is next
FIB-SEM is a powerful tool, particularly for a cellular system that is as unusual as malaria parasites, as it allows determination of general ultrastructural organization and basic measurements to cover knowledge gaps that in other model eukaryotes have long been closed. In this study, we applied volumetric electron microscopy to the transmissible gametocyte stages of the malaria parasite. This allowed us to place findings from the past into their current molecular context and tackle questions that are challenging to address using single sections or serial sections with low z-resolution such as the connectivity within or between various organelles. Furthermore, these data allowed us to create the first 3D visualization of Golgi, gametocyte cytostome, Garnham body, and ER in gametocytes, improving our understanding of the cellular architecture in this part of the life cycle. In doing so, we are aware of the descriptive nature of this study and the limitations of the tool applied. Any of the hypotheses put forward are meant to provide a foundation for controlled molecular studies and targeted approaches that are more suitable to identify the underlying molecular players.

## Methods
### Parasite culture & gametocyte induction
All parasite material used in this study is derived from the NF54/iGP2 strain. NF54/iGP2 parasites were previously shown to be able successfully complete the whole life cycle as well as not markedly differ from wild-type NF54 parasites in other parameters that were investigated[97]. This parasite strain has the desirable property that gametocytogenesis can be selectively induced by removal of glucosamine from the medium, which triggers sexual commitment through the transcriptional cascade centred around the AP2G gene[106]. Asexual blood-stage parasites (ABS) were maintained according to standard culturing procedure in RPMI [7.4] supplemented with 10% human serum and 5% haematocrit using standard culturing technique in a semi-automatic culturing system[9,107]. For the maintenance of the NF54/iGP2 strain, an additional supplement of 2.5 mM D-(+)-glucosamine hydrochloride (Sigma #1514) was used. To induce gametocytogenesis, glucosamine was omitted from the culturing medium. Between days 4 and 8 after gametocyte induction 50 mM N-acetylglucosamine (Sigma #A6525) was used to eliminate ABS parasites. On day 14 gametocyte-infected RBCs were separated from uninfected RBCs through magnetic separation using standard methods[108]. Close attention was paid to prewarm and maintain all solutions and apparatus at 37 °C to avoid unintended activation of mature gametocytes. As reference material trophozoite and schizont stages from mixed ABS cultures were similarly enriched through magnetic separation and processed alongside

the gametocyte cultures. For sample 3, N-acetylglucosamine treatment was omitted to control for phenotypes caused by this treatment. This omission leads to presence of ABS in the resulting images (sample 3).

### Sample preparation for electron microscopy
Samples (Table 1) were prepared as described previously[25]. Briefly, the enriched infected red blood cells were fixed using 2% glutaraldehyde in 0.1 M cacodylate (pH 7.4) buffer overnight at a temperature of 4 °C. The fixed cells were then washed and the cell-pellet was resuspended in 3% ultra-low-gelling agarose, solidified, and cut into small blocks. The agarose blocks containing the fixed cells were postfixed for 1 h at room temperature using a solution of 2% osmium tetroxide and 1.5% potassium ferrocyanide in 0.1 M cacodylate buffer containing 2 mM CaCl$_2$, washed in MQ and then treated with 0.5% thiocarbohydrazide for 30 min at room temperature. After washing, the agarose blocks were again suspended in 2% osmium for 30 min at room temperature, washed, and then placed in a 2% aqueous uranyl acetate solution overnight at 4 °C. The blocks were then washed and placed in a lead aspartate solution (pH 5.5) for 30 min at 60 °C, washed, dehydrated using an ascending series of aqueous ethanol solutions, and subsequently transferred via a mixture of acetone and Durcupan to pure Durcupan (Sigma) as an embedding medium. The aforementioned staining procedure is primarily optimized for high membrane contrast and staining of lipid-rich structures due by utilizing the combination of potassium ferrocyanide, osmium tetroxide and thiocarbohydrazide[109]. Uranyl acetate and lead aspartate furthermore stain DNA/RNA, proteins as well as carbohydrates[110,111].

### FIB-SEM
After polymerization and in order to create a smooth plane for FIB-SEM imaging, the sample surface was smoothened using an ultra-microtome (Reichert Ultracut S ultramicrotome (Leica microsystems)). The region of interest was cut out of the resin block and glued on an SEM stub with carbon tape, using conductive silver paint. The samples were then coated with a gold sputter coater (Edwards, Stockholm, Sweden) before introduction into a Zeiss Crossbeam 550 FIB-SEM (Carl Zeiss).

Multiple coarse trenches were milled using a 30 kV@30 nA probe to choose the regions of interest for further 3D volume imaging. Parameters for serial sectioning imaging of large regions were set using the Atlas 3D software (Atlas Engine v5.3.3). The large trenches were first smoothened using a 30 kV@1.5 nA FIB probe, and thereafter a 30 kV@700pA probe current was used for serial FIB milling. InLens secondary and backscattered electron microscopy images were simultaneously collected at an acceleration voltage of 2.0 kV with a probe current of 500 pA. The backscattered grid was set to −902 V. For noise reduction, images were acquired using line average ($n = 1$) and a dwell time of 3.0 μs. The milling and imaging processes were continuously repeated and long series of images were acquired.

### Serial sectioning
To have a lower noise reference and identify potential FIB imaging artefacts one gametocyte sample was also analysed through serial sectioning combined with SEM. After checking for desired density of cells with toluidine blue corresponding resin block was trimmed to the

Article

desired size and mounted onto the chuck of a Leica Artos 3D Ultra-microtome. The block was sectioned into thin 80 nm slices and sections were collected onto Indium Tin Oxide glass coated with 5 nm carbon. The sections were then imaged with a scanning electron microscope (Sigma300, Zeiss) at an acceleration voltage of 30 kV (HDBSD, 60 μm, high current) using Atlas 5 software. In total 47 consecutive images were taken leading to a total imaged volume of 15124 um³.

## Post-processing and segmentation

All processing, visualization and analysis performed in the ORS Dragonfly software (V2022.2). Wherever necessary image stacks were aligned using the mutual information and sum of squared differences registration method and results were manually controlled and adjusted whenever necessary. Contrast was enhanced through application of contrast limited adaptive histogram equalization (CLAHE). 3D segmentation was performed using either manual segmentation or deep learning based segmentation, based on which cellular feature was segmented. Deep-learning based segmentations were controlled for errors and manually adjusted when necessary. For 3D rendering the segmented regions of interest were converted to triangle meshes.

## Comparative analysis of osmiophilic bodies and mitochondria

Individual mitochondria or osmiophilic bodies (OBs) were isolated from the segmentation masks with the 6-connected criterion as implemented in ORS Dragonfly. Objects with a voxel count < 375 (0.1 um³) were discarded as false positive assignments from the automated segmentation due to their implausibly small size. Generating accurate OB counts for female gametocytes proved challenging for our analysis pipeline as OBs were sporadically densely packed which led to multiple adjacent OBs being counted as a single OB. To circumvent this skewing average size and shape measurements in female gametocytes, we filtered outliers (Z-score > 3) for downstream analysis. The scalar generator in ORS Dragonfly was used to extract shape and size measurements for the individual objects. Significance of observed differences between male and female gametocytes was tested with a two-tailed t-test.

## Immunofluorescence assays

To induce gametocytes, NF54 parasites were swapped from their earlier described complete culture media to RPMI media supplemented with 0.5% Albumax (AlbuMAX II™, Thermo Fisher, #11021-037) at 26 h post invasion. The parasites were allowed to commit for 36 h, before being put back on complete culture media. At day 12 post induction, stage V gametocytes were stained in 100 nM MitoTracker™ (MitoTracker™ Orange CMTMRos, Thermo Fisher, #M7510) diluted in complete media at 37 °C for 30 min. Stage V gametocyte samples were diluted 1:10 in warm complete media before being allowed to settle on pre-heated Poly-L-Lysine coated coverslips (Corning, #354085) for 15 min at 37 °C, while activated gamete samples were mixed 1:1 with in 30 μM xanthurenic acid (Sigma Aldrich, #D120804) and subsequently settled for 15 min at RT. Both samples were fixed with 4% paraformaldehyde (Thermo Fisher, #28906)) and 0.0075% glutaraldehyde (Panreac, #A0589,0010) in PBS for 20 min at RT. The fixed samples were, permeabilized with 0.1% Triton X-100 in PBS for 10 min and blocked in 3% BSA (Sigma Aldrich, #A9418) for 1 h at RT. To differentiate between male and female gametocytes, the samples were stained with 1:500 primary α-Tubulin antibody (Thermo Fisher, #MA1-19162) for 1 h at RT, which was visualised with 1:500 Donkey anti-Mouse Alexa Fluor™ 647 (Thermo Fisher, #A-31571). Finally, samples were stained with 300 nM DAPI (Thermo Fisher, #62248) for 30 min at RT before being mounted on a microscope slide using VECTASHIELD (VWR, #H-1000) and sealed in nail polish. The samples were imaged on an confocal LSM900 microscope with airyscan (Zeiss). Z-stack images were obtained with a 0.14 μm step size. Laser power and detector sensibility are maintained

the same throughout all the images. A maximum projection was created for every image, in which the tubulin signal was pre-set in order to differentiate between male and female gametocytes.

## Reporting summary

Further information on research design is available in the Nature Portfolio Reporting Summary linked to this article.

## Data availability

All FIB-SEM data, both raw image stacks and individual analysed cells, as well as the serial sectioning data used this study have been deposited on the Electron Microscopy Public Image Archive and can be retrieved under accession code EMPIAR-12160. Data underlying graphs and quantitative analysis are provided in the Source Data File. Source data are provided with this paper.

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

## Acknowledgements

We thank the members of the Molecular and Cellular Parasitology team for fruitful discussions. We also thank Sabrina Absalon for critically proofreading the manuscript. F.E. and T.W.A.K. were supported by the Netherlands Organisation for Scientific Research (NWO-VIDI 864.13.009), C.B. by a PhD. fellowship from the Radboud Institute for Molecular Life Sciences, Radboudumc (RIMLS018-009b), J.M.J.V. was supported by an individual Radboudumc Master-PhD grant. R.R., N.S., and A.A. are supported by an ERC Advanced Investigator grant (H2020-ERC-2017-ADV-788982-COLMIN for Nico Sommerdijk). A.A. is also supported by the NWO (VI.Veni.192.094).

## Author contributions

F.E., R.R., C.B., M.K.L., and J.M.J.V. performed experiments. R.E.S. provided conceptual advice and analysis. F.E. analysed the results, prepared illustrations and wrote the manuscript draft. A.A. provided resources technical expertise and conceptual advice. N.S. provided resources and conceptual advice. T.W.A.K. contributed to conceptual development, provided resources, and edited the manuscript. All authors contributed to data interpretation and provided feedback on the manuscript. All authors approved the final version of the manuscript.

## Competing interests

The authors declare no competing interests.
