## [Transparent Peer Review file · Nature Communications]

Comparative 3D ultrastructure of *Plasmodium falciparum* gametocytes

Corresponding Author: Dr Taco Kooij

Version 0:

Reviewer comments:

Reviewer #1

(Remarks to the Author)

Review of Kooij Nat Comm 2023 – 3D structure of gams

The manuscript from Evers et al. uses multiple microscopy techniques to evaluate the 3D structure of *Plasmodium falciparum* gametocytes. The studies verify some previous findings and introduce a series of new findings about the organelles within gametocytes.

The manuscript is certainly interesting and provides some examples of very high-quality microscopy. The major weakness of the manuscript is that it is entirely descriptive. The findings are likely real, but they would be more convincing if there was some quantification of the findings (see major comment 3 below).

MAJOR

1. In Fig 1A, 1B, and 1C, what specific features are used to identify the Golgi by electron microscopy? Similarly, what criteria are used to distinguish mitochondria from apicoplasts? This is difficult to differentiate from the sample images shown. This is discussed some (lines 246-248), but should be explained fully and done earlier when it is first mentioned.
2. Figure 2 needs a color-coded legend. Even though it is present in Figure 1, it should be shown again in each figure so that the reader does not need to go back and forth to understand what the colors mean.
3. The findings from the volume electron microscopy studies should be quantified wherever possible. How many gametocytes were imaged? For example, the number of OBs with “tails” should be quantified in male and female gametocytes. In how many gametocytes were the extraparasitic structures observed? This is especially true when discussing the possible connections between the ER and IMC in the gametocytes. The authors are proposing a novel method of protein export. This finding is really important but requires quantification to be more believable.
4. The discussion of internal structures within the mitochondria is interesting. What is the formal difference between these structures in ABS mitochondria and “cristae”? I had to read the mitochondria section multiple times to understand it. It would be helpful to give a few extra sentences in this section to make it more clear. The separation of individual mitochondria is interesting and should be made stronger by some quantitative data. Is the separation seen in a large percentage of FIB-SEM renderings of the gametocytes? The fluorescence data is difficult to interpret because the MitoTracker staining may just be uneven.

MINOR

1. The introduction paragraph is very nice but should include some additional references for the statements about “diverged very early in eukaryotic evolution” and “organelle biology has been shown to deviate from standard”.
2. Line 123 (line 162, and throughout). I think that “sex” is the correct term and not “gender” for gametocytes. The sentence should be “no evidence for sex specificity”.
3. The text is dense and, at times, hard to follow. If possible, some care should be given to making the text somewhat easier to follow. This could be done by including some additional sentences with clearer explanations. I know this is difficult, but it is worth a concerted effort.

Reviewer #2

(Remarks to the Author)

The manuscript by Evers et al. describes the analysis of the gametocytic stage of *Plasmodium falciparum* by FIB/SEM. This is a technically sound and carefully conducted study that provides some previously unseen details of the stage of the parasite that propagates the life cycle through sexual reproduction in the vector. The study pays attention to individual organelles, their abundance and points of contacts, as well as differences in overall cell structure compared to asexual blood stages.

Major discoveries include the description of multiple discrete mitochondria, a morphologically unique cytostome, and the observation of ER tubules as a major interacting element.

Taken as a whole, this is a fine work that definitely advances the cell biology of the malaria parasite and provides a basis for targeted functional studies. However, it is a purely descriptive work that does not bring any new insight into the biology of the parasite and "only" refines already known data. A few morphological novelties such as the number of mitochondria or the morphology of the cytostome are, in my opinion, not sufficient for publication in a high profile journal such as *Nature Communications*.

Version 1:

Reviewer comments:

Reviewer #1

(Remarks to the Author)

The manuscript provides a high resolution evaluation of the morphologic features of *Plasmodium falciparum* gametocytes. For 3D reconstructions, this is the highest resolution analysis thus far, relying mostly of FIB-SEM and machine learning segmentation for segmentation. In the first submission, the findings were entirely descriptive. In the revised submission, the authors have greatly improved the manuscript by providing some quantitative data to support their conclusions. In summary, the authors have almost entirely addressed my previous concerns.

This manuscript validates some previously known findings. More importantly, the authors have identified multiple interesting and novel morphologic features in gametocytes. These include some sex-differences in male vs. female gametocytes. Separately, the authors convincingly demonstrate the presence of multiple mitochondria within gametocytes. This is an exciting finding.

I would encourage the authors to use larger fonts in their figures. However, this is minor and only a matter of style.

I have no additional suggested changes.

Reviewer #3

(Remarks to the Author)

The authors have done a nice job incorporating new data, quantitative analysis, and expanded discussion of posited hypotheses. The figures are clearer and the color-coded annotations and cartoon in Fig 4 are particularly helpful. The manuscript overall is much improved and is suitable for publication in its current form.

Summary

We would like to thank the editor and all the reviewers all for the time and effort that you dedicated to review our manuscript as well as all the constructive criticism and suggestions that helped improve quality, readability and merit of this work for the wider community. In an effort to ensure both reproducibility and identify more mature male gametocytes to be able to more accurately quantify sexual dimorphisms, we generated, imaged and analyzed another sample containing a mixture of asexual and sexual blood stages. Improvements in the acquisition method allowed this biological replicate to have a larger field of view and much longer acquisition times, leading to a large increase in the amount of fully imaged cells. Reassuringly, our previous observations are consistent with the new data. The male mature gametocytes desired for quantification and a corresponding amount of mature female gametocytes (to minimize bias caused by having most male gametocytes from one sample) were obtained. While a complete analysis of this enormous dataset was not feasible within a reasonable time frame, these new data are deposited alongside the initial dataset in the EMPIAR database and will be available to the public in their entirety. Furthermore, we expanded our analysis to also include quantitative analysis of sexual dimorphisms of the osmiophilic bodies as well as add quantitative analysis to the observation of multiple mitochondria in gametocytes. Due to the inclusion of more male gametocytes, this allowed us to detect another sexual dimorphism in regard to mitochondrial volume as well as morphology. These new insights prompted us to expand on and redistribute figures between main text and supplement. The new data and quantifications of osmiophilic bodies are reflected in a new main text figure (Figure 2) while Figure 6 was adapted to now include quantifications of mitochondrial morphology and number and corresponding renderings in Figure S11. To keep overall figure number consistent, observations on putative organelle interfaces (Figure 6) were moved to the supplement (Figure S12). Similarly, a new subsection on osmiophilic bodies was added to the manuscript and the subsection “Multiple mitochondria in disguise” was expanded to include the new quantifications. Taken together we added a new dataset to increase the number of previously rare cell populations and to validate our previous observations. Furthermore, we included quantifications where useful to supplement our previously qualitative observations.

Data accessibility

The underlying data are deposited at the **Electron Microscopy Public Image Archive (EMPIAR)** under the entry EMPIAR-12160. They can be accessed by the reviewers with view-only permissions and the option to download the data. For this the following credentials should be used to log in to the deposition system of EMPIAR (<https://www.ebi.ac.uk/empiar/deposition/login/?next=/empiar/deposition/>)
username: **review_cf76eb8b**
password: **6feef371**

Reviewer #1 (Remarks to the Author):

Review of Kooij Nat Comm 2023 – 3D structure of gams

The manuscript from Evers et al. uses multiple microscopy techniques to evaluate the 3D structure of *Plasmodium falciparum* gametocytes. The studies verify some previous findings and introduce a series of new findings about the organelles within gametocytes.

The manuscript is certainly interesting and provides some examples of very high-quality microscopy. The major weakness of the manuscript is that it is entirely descriptive. The findings are likely real, but they would be more convincing if there was some quantification of the findings (see major comment 3 below).

We thank the reviewer for the appraisal of the quality of our work. We have expanded on our previous submission supporting the robustness of our data and including additional quantitative measures to further increase the significance of our studies.

MAJOR

1. In Fig 1A, 1B, and 1C, what specific features are used to identify the Golgi by electron microscopy? Similarly, what criteria are used to distinguish mitochondria from apicoplasts? This is difficult to differentiate from the sample images shown. This is discussed some (lines 246-248), but should be explained fully and done earlier when it is first mentioned.

In order to identify the Golgi and distinguish mitochondria from apicoplast as well as all other organelle assignments, we adhered to the characteristics described in classical electron microscopy¹⁻⁴. Like the reviewer we also believe that the reader should be equipped with the tools to identify the organelles within the exemplar micrographs without having to consult other manuscripts or books. We chose to place these exact descriptions of the features in the respective subsections discussing these organelles as these are the points where the reader is most likely to encounter figures with respective micrographs and this avoids 'cluttering' the initial section introducing the overall morphology of the different cells and salient distinguishing features. We do agree with the reviewer that it is important for the reader to know that organelle assignments were made within an existing framework and in agreement with previous EM works when the structures are encountered first and as such we included the following sentence in the introductory paragraph to reflect that:

L103-105: *"Appearance of nucleus, ER, Golgi, mitochondrion and apicoplast matched that found in classical works¹⁻⁴ and allowed us to confidently assign and reconstruct these organelles in 3D."*

The more detailed descriptions can still be found in the respective subsections.

Golgi/ER (L289-291): *"This coincides with the distribution of a closely associated organelle, the Golgi, which, in Plasmodium, takes the rudimentary form of dispersed unstacked cisternae and is recognizable as a smooth membraned vesicle cluster^{2, 5-7"}*

Mitochondrion/Apicoplast (L446-448): *"The apicoplast is recognizable as a clearly distinct structure from the mitochondrion due to the thicker appearance of its four surrounding membranes and lack of internal membranous structures and more electron lucent lumen."*

Below are two exemplary micrographs with legends highlighting differences between aforementioned organelles. These structures are also differentially labelled in the exemplary micrographs in Figure 1 (Golgi/ER, Mitochondrion/Apicoplast) and Figure 5 (Mitochondrion/Apicoplast)

2. Figure 2 needs a color-coded legend. Even though it is present in Figure 1, it should be shown again in each figure so that the reader does not need to go back and forth to understand what the colors mean.

Figure two uses colors differently than Figure 1. Grey with different opacity was used to give the context of the outline of the red blood cell (highly transparent outline) and parasite within (opaque grey outline). The extraparasitic structures were not assigned to different classes but instead each connected structure was assigned to 32-color lookup table to make the individual exported units more distinguishable in the 2D projection. As such a legend like the one given in Figure 1 is not possible as no distinct classes are signified by the colors. We do understand the potential for confusion and to reflect that we added this snippet to signify the meaning of the different colors *“extraparasitic structures in arbitrary colors to distinguish separate and connected extraparasitic structures”*

3. The findings from the volume electron microscopy studies should be quantified wherever possible. How many gametocytes were imaged? For example, the number of OBs with “tails” should be quantified in male and female gametocytes. In how many gametocytes were the extraparasitic structures observed? This is especially true when discussing the possible connections between the ER and IMC in the gametocytes. The authors are proposing a novel method of protein export. This finding is really important but requires quantification to be more believable.

OB: We agree with the reviewer that the OB observations deserve more rigorous and quantitative underpinning and as such we extended our dataset to include more male cells and applied a more

quantitative analysis with a distinctly specified quantity of cells. With this increase in analysis, we updated the findings accordingly and moved this part of the manuscript to a specific subsection instead of being contained within the general morphology section as previously, and dedicated a main text figure to this new analysis. In general, we find large and significant differences in OB number, volume and particle size while we find small but significant differences in aspect ratio and sphericity of the particles (Fig. 2A-J). While generally the findings hold, the more rigorous analysis revealed that the tail-like forms are more prevalent in females than initially thought, as they are less distinct and further occluded by the close vicinity in which we often find them. While still comparatively more prevalent in males, we toned down our language accordingly. Furthermore, we created ten 3D-renderings (Fig. 2K) to give a sense of spatial distribution and summarized differences between the two sexes.

Extrarasitic structures: We changed the text to reflect that extrarasitic structures are present in all gametocyte-infected RBCs but are less numerous than in ABS.

ER/IMC: Unfortunately, with our current analysis methods we do not have accurate or specific masks for the IMC, our evidence stems from manual inspection of stacks and observation of membrane continuity. As a result our observations remain qualitative. To give an indication of frequency of these observations, we changed the text to reflect that the ER/IMC association is a universal feature found in all gametocytes that we have imaged with an IMC. In the text, we emphasize that this is anecdotal evidence and while we give qualifiers like “frequent” “less frequent” to indicate relative abundance, we think this section should be more understood as a collection of the different interactions and their speculative function rather than a systematic analysis. To not overstate the importance of these findings, we have moved the corresponding figure to the supplemental information.

4. The discussion of internal structures within the mitochondria is interesting. What is the formal difference between these structures in ABS mitochondria and “cristae”? I had to read the mitochondria section multiple times to understand it. It would be helpful to give a few extra sentences in this section to make it more clear. The separation of individual mitochondria is interesting and should be made stronger by some quantitative data. Is the separation seen in a large percentage of FIB-SEM renderings of the gametocytes? The fluorescence data is difficult to interpret because the MitoTracker staining may just be uneven.

Fluorescence data: We agree that, on its own, the fluorescence data are difficult to interpret as we discuss in the manuscript. Here however, the purpose is to cross-validate our findings, as it is unlikely that two techniques based on different principles for visualization produce the exact same artifacts. It is also there to derive how the notion of a single mitochondrion came to be, as fluorescence data were and remain the main modality to shape our view of the morphology of the mitochondrion.

EDMG/Cristae (L339-341): To further clarify the different structures we added the following sentence. *“EDMGs are clearly distinct from the cristae observed in gametocytes based on their homogenous electron dense appearance without clear membrane delimitation as opposed to the cristae which are a membrane that surrounds an electron lucent lumen.”*

Multiple mitochondria: The appearance of multiple separate mitochondria is a feature observed universally in all early and late-stage gametocytes of both sexes. We have now included a further quantification of individual mitochondria and the sexual dimorphism of this organelle which was not evident from our previous analysis.

MINOR

1. The introduction paragraph is very nice but should include some additional references for the statements about “diverged very early in eukaryotic evolution” and “organelle biology has been shown to deviate from standard”.

The following citations were added to support these claims:

Apicomplexan parasites diverged early in eukaryotic evolution

1. Escalante, A.A. & Ayala, F.J. Evolutionary origin of Plasmodium and other Apicomplexa based on rRNA genes. *Proceedings of the National Academy of Sciences* 92, 5793-5797 (1995).

2. Templeton, T.J. et al. Comparative analysis of apicomplexa and genomic diversity in eukaryotes. *Genome Res* 14, 1686-1695 (2004).

3. Burki, F., Roger, A.J., Brown, M.W. & Simpson, A.G.B. The New Tree of Eukaryotes. *Trends Ecol Evol* 35, 43-55 (2020).

Apicomplexan organelle biology is divergent

1. Klinger, C.M., Nisbet, R.E., Ouologuem, D.T., Roos, D.S. & Dacks, J.B. Cryptic organelle homology in apicomplexan parasites: insights from evolutionary cell biology. *Current Opinion in Microbiology* 16, 424-431 (2013).

2. Koreny, L. et al. Stable endocytic structures navigate the complex pellicle of apicomplexan parasites. *Nature Communications* 14, 2167 (2023).

2. Line 123 (line 162, and throughout). I think that “sex” is the correct term and not “gender” for gametocytes. The sentence should be “no evidence for sex specificity”.

All mentions of gender have been changed to sex.

3. The text is dense and, at times, hard to follow. If possible, some care should be given to making the text somewhat easier to follow. This could be done by including some additional sentences with clearer explanations. I know this is difficult, but it is worth a concerted effort.

We prefer a concise and consequently dense writing style for several reasons: 1) it brings across the main messages more effectively, 2) not to make an overly long manuscript, and 3) much can be readily interpreted from close observation of the imaging data. In several places, we have nevertheless expanded our reasoning and explanations a little for better understanding.

Reviewer #2 (Remarks to the Author):

The manuscript by Evers et al. describes the analysis of the gametocytic stage of *Plasmodium falciparum* by FIB/SEM. This is a technically sound and carefully conducted study that provides some previously unseen details of the stage of the parasite that propagates the life cycle through sexual reproduction in the vector. The study pays attention to individual organelles, their abundance and points of contacts, as well as differences in overall cell structure compared to asexual blood stages.

Major discoveries include the description of multiple discrete mitochondria, a morphologically unique cytotome, and the observation of ER tubules as a major interacting element.

Taken as a whole, this is a fine work that definitely advances the cell biology of the malaria parasite and provides a basis for targeted functional studies. However, it is a purely descriptive work that does not bring any new insight into the biology of the parasite and "only" refines already known data. A few morphological novelties such as the number of mitochondria or the morphology of the cytotome are, in my opinion, not sufficient for publication in a high profile journal such as *Nature Communications*.

We thank the reviewer for the appraisal of the quality of our work and the recognition of the novelty of several of our findings. While we do agree that our study does not provide insights in molecular mechanisms underlying our observations, the manuscript does contain cross-validation by different imaging modalities thus providing new insights into the biology of the parasite, not just refining our understanding but also refuting some long-standing dogma's. By including further quantitative analyses these observations have been further substantiated in this new version of the paper.

Reviewer #3 (Remarks to the Authors):

In this study, the authors have used focused ion beam scanning electron microscopy, serial section electron microscopy and immunofluorescence microscopy of fixed *Plasmodium falciparum* cells to visualize the three dimensional morphology and ultrastructure of gametocyte stage *P. falciparum* parasites. The resulting dataset provides insight into the organellar organization in this life cycle stage that is important in malaria transmission. As the authors have themselves acknowledged, this work is wholly descriptive in nature and intended to provide a foundation for controlled molecular studies and targeted approaches that are more suitable for identifying the underlying molecular players. However, a more detailed discussion of what these controlled molecular studies might be, and how these studies could realistically lead to the discovery of new potential drug targets would help to clarify the impact of this work.

Major points:

A discussion of the potential implications of the various observations and corresponding hypotheses suggested throughout the work would help readers to better judge the impact of the work, particularly with regard to direct ways in which this work could facilitate the discovery of novel drug targets and the development of new therapies, as this was presented in the introduction as the primary motivation for the study. What experiments would need to be done to test the hypotheses generated from the observations shown here? If successful, how would the results of these further experiments translate into discovery or validation of new drug targets? Which of these observations represent points of weakness or differences in basic biology that could be exploited for developing new treatments? What are the broader implications of each of these observations in terms of building a better understanding of parasite biology?

We have added hypotheses regarding drugability, parasite biology and future experiments for the cytotome and the multiple mitochondria. Changes are spread throughout the manuscript and highlighted in blue.

Minor points:

1. Given the broad range of length scales achievable by the increasingly varied and powerful techniques beautifully summarized by the authors in the introduction, “high resolution” is a vague and relative descriptor for the data shown here. It would be good to replace this term with a more specific descriptor that gives the reader an accurate sense of the resolution range of this work – ie, sub-micron, nanometer, sub-nanometer, near-atomic, atomic.

We agree and have changed to more specific descriptors.

L32 nanometer scale

L77 nanometer

2. Including the legend indicating the colors corresponding with various ultrastructural features in all figures would be helpful, so that the reader does not have to keep referring back to the legend in Figure 1.

Legends to map colors to structures have been added to figures where relevant.

3. In Figure 3, if it would be possible to adjust the slices shown so that the invagination of the PVM and PPM and the continuity between the cytosome lumen and the RBC cytosol was equally visible in all examples shown, that would make this figure easier to interpret.

Unfortunately, this is not feasible for all examples. These images are from serial sectioning data with 80nm z-steps, so the jumps are relatively large compared to the FIB-SEM data. Also showing the invagination site often precludes us from seeing the features defining this subtype of cytosome. The representative frames were chosen to show both continuity and defining internal features wherever possible and prioritized to show the differentiating features wherever both are not possible. The underlying ssSEM data are deposited on EMPIAR alongside the FIB-SEM data.

4. More granular labeling of the panels in Figure 3 with additional corresponding references to individual Fig 3 panels in the main text, as well as arrowheads on the images indicating each of the defining features described in the figure legend would also help the reader to better interpret the figure

Figure 3 was updated to now include (color-coded) symbols for each of the defining features overlaid on the micrographs.

5. Additionally, in Figure 3, a cartoon schematic of the three categories of putative gametocyte cytosomes, demonstrating the defining features described in the text, would be helpful.

A cartoon schematic to better illustrate the salient features of the observed morphology was added to Figure 3.

Literature

1. Sinden, R.E., Canning, E.U., Bray, R.S., Smalley, M.E. & Garnham, P.C.C. Gametocyte and gamete development in *Plasmodium falciparum*. *Proceedings of the Royal Society of London. Series B. Biological Sciences* **201**, 375-399 (1978).
2. Sinden, R.E. Gametocytogenesis of *Plasmodium falciparum* in vitro: an electron microscopic study. *Parasitology* **84**, 1-11 (1982).
3. Ponnudurai, T., Lensen, A.H.W., Meis, J.F.G.M. & Meuwissen, J.H.E.T. Synchronization of *Plasmodium falciparum* gametocytes using an automated suspension culture system. *Parasitology* **93**, 263-274 (1986).
4. Aikawa, M., Huff, C.G. & Sprinz, H. Comparative fine structure study of the gametocytes of avian, reptilian, and mammalian malarial parasites. *Journal of Ultrastructure Research* **26**, 316-331 (1969).

5. Adisa, A. et al. Re-assessing the locations of components of the classical vesicle-mediated trafficking machinery in transfected *Plasmodium falciparum*. *International journal for parasitology* **37**, 1127-1141 (2007).
6. Krai, P., Dalal, S. & Klemba, M. Evidence for a Golgi-to-Endosome Protein Sorting Pathway in *Plasmodium falciparum*. *PLOS ONE* **9**, e89771 (2014).
7. Hallée, S. et al. Identification of a Golgi apparatus protein complex important for the asexual erythrocytic cycle of the malaria parasite *Plasmodium falciparum*. *Cell Microbiol* **20**, e12843 (2018).

Response to reviewers

Once more we would like to thank the editor and all the reviewers for the time and effort that you significantly improved our manuscript. We have carefully gone through the author checklist amending the manuscript where relevant and responding accordingly in the form. As the reviewers had no further suggestions for improvement of the manuscript there is no need for a point-by-point response. We would like to use the opportunity to thank the reviewers once more for their positive and constructive feedback on our manuscript.

Reviewer #1 (Remarks to the Author):

The manuscript provides a high resolution evaluation of the morphologic features of *Plasmodium falciparum* gametocytes. For 3D reconstructions, this is the highest resolution analysis thus far, relying mostly of FIB-SEM and machine learning segmentation for segmentation. In the first submission, the findings were entirely descriptive. In the revised submission, the authors have greatly improved the manuscript by providing some quantitative data to support their conclusions. In summary, the authors have almost entirely addressed my previous concerns.

This manuscript validates some previously known findings. More importantly, the authors have identified multiple interesting and novel morphologic features in gametocytes. These include some sex-differences in male vs. female gametocytes. Separately, the authors convincingly demonstrate the presence of multiple mitochondria within gametocytes. This is an exciting finding.

I would encourage the authors to use larger fonts in their figures. However, this is minor and only a matter of style.

We have increased the font size slightly wherever we saw fit and feasible. We are open to amend figures further at the editor's discretion.

I have no additional suggested changes.

Reviewer #3 (Remarks to the Author):

The authors have done a nice job incorporating new data, quantitative analysis, and expanded discussion of posited hypotheses. The figures are clearer and the color-coded annotations and cartoon in Fig 4 are particularly helpful. The manuscript overall is much improved and is suitable for publication in its current form.